# Rapid fabrication of reinforced and cell-laden vascular grafts structurally inspired by human coronary arteries

Tamara L. Akentjew[1,2,3,4], Claudia Terraza[1,2], Cristian Suazo[1,2], Jekaterina Maksimcuka[5], Camila A. Wilkens[1,2,3], Francisco Vargas[6], Gabriela Zavala[1,2], Macarena Ocaña[1,2], Javier Enrione[7], Claudio M. García-Herrera [8], Loreto M. Valenzuela[4,9,10], Jonny J. Blaker [11], Maroun Khoury[1,2,3] & Juan Pablo Acevedo[1,2,3]

Design strategies for small diameter vascular grafts are converging toward native-inspired tissue engineered grafts. A new automated technology is presented that combines a dip-spinning methodology for depositioning concentric cell-laden hydrogel layers, with an adapted solution blow spinning (SBS) device for intercalated placement of aligned reinforcement nanofibres. This additive manufacture approach allows the assembly of bio-inspired structural configurations of concentric cell patterns with fibres at specific angles and wavy arrangements. The middle and outer layers were tuned to structurally mimic the media and adventitia layers of native arteries, enabling the fabrication of small bore grafts that exhibit the J-shape mechanical response and compliance of human coronary arteries. This scalable automated system can fabricate cellularized multilayer grafts within 30 min. Grafts were evaluated by hemocompatibility studies and a preliminary in vivo carotid rabbit model. The dip-spinning-SBS technology generates constructs with native mechanical properties and cell-derived biological activities, critical for clinical bypass applications.

[1] Laboratory of Nano-Regenerative Medicine, Faculty of Medicine, Universidad de los Andes, San Carlos de Apoquindo 2200, Las Condes, Santiago 7620001, Chile. [2] Cells for Cells, Avda. Plaza 2501, Las Condes, Santiago 7620157, Chile. [3] Consorcio Regenero, Avda. Plaza 2501, Las Condes, Santiago 7620157, Chile. [4] Department of Chemical and Bioprocess Engineering, School of Engineering, Pontificia Universidad Católica de Chile, Avda. Vicuña Mackenna 4860, Macul, Santiago 7820436, Chile. [5] School of Materials, MSS Tower, The University of Manchester, Manchester M13 9PL, UK. [6] Departamento de Cirugía Vascular y Endovascular, Pontificia Universidad Católica de Chile, Avda. Libertador Bernando O'Higgins 340, Santiago 8331150, Chile. [7] Biopolymer Research and Engineering Lab (BiopREL), School of Nutrition and Dietetics, Faculty of Medicine, Universidad de los Andes, Avda. Plaza 2501, Las Condes, Santiago 7620157, Chile. [8] Departmento de Ingeniería Mecánica, Universidad de Santiago de Chile, Avda. Libertador Bernando O'Higgins 3363, Estación Central, Santiago 9170022, Chile. [9] Institute for Biological and Medical Engineering, Schools of Engineering, Medicine and Biological Sciences, Pontificia Universidad Católica de Chile, Libertador Bernando O'Higgins 340, Macul, Santiago 7820436, Chile. [10] Center of Nanotechnology Research and Advanced Materials "CIEN -UC", Pontificia Universidad Católica de Chile, Avda. Libertador Bernando O'Higgins 340, Macul, Santiago 7820436, Chile. [11] Bio-Active Materials Group, School of Materials, MSS Tower, The University of Manchester, Manchester M13 9PL, UK. Correspondence and requests for materials should be addressed to J.P.A. (email: jpacevedo@uandes.cl)

Coronary heart disease is the main cardiovascular disease, and due to an increasing aging population[1], global deaths attributed to this cause are expected to reach 22.2 million in 2020[2]. Although autologous vessels remain the best alternative for bypass surgeries, their application is limited by pre-existing vascular diseases, previous autograft harvest, limited length, low quality and considerable morbidity associated with autologous harvesting[3–5].

Synthetic vascular prostheses made of expanded polytetrafluoroethylene (ePTFE) or poly(ethylene terephthalate) have been widely used for large diameter blood vessel replacement[6]. However, when used as small diameter, synthetic prostheses are rapidly occluded due to thrombosis and intimal hyperplasia, mainly caused by blood–graft interface contact activation[7] and compliance mismatch between natural blood vessels and the graft[8]. This renders synthetic prostheses unsuitable for replacement or bypass surgeries of small diameter blood vessels such as coronary arteries (ID <6 mm).

Tissue-engineered vascular grafts are emerging as alternatives to this problem, and seek to mimic the structural arrangement and mechanical response of collagen/elastin fibres to sustain physiological pressures and provide the necessary compliance and recoil[3].

Differences in fibre orientation and quantity in each of the three artery layers (intima, media and adventitia) govern the mechanical properties of vessels in the vascular system[9,10]. Studies focused on numerical modelling of human coronary artery mechanics and microscopy have determined that collagen fibres in the adventitia are bi-directionally oriented at angles of +/−67° relative to the circumferential axis (see Fig. 1), while those in the media are oriented at +/−21[11,12]. Arteries subjected to internal pressure oscillation exhibit a non-linear elastic response, with a characteristic J-shaped stress–strain deformation profile[13]. When the pressure increases beyond the elastic resistance of elastin fibres, wavy collagen fibres stretch and start governing the mechanical behaviour, attributed to the linear section of the stress–strain curve[14].

Mismatch of mechanical response between the prosthesis and native vessel can lead to hyperplasic intimal formation at the anastomosis zone following changes in internal pressure[15].

Several studies have reported new strategies for the fabrication of vascular grafts using natural, synthetic or hybrid materials[8], such as direct decellularization[16] of xenogeneic vessels[17], self-assembled cell sheets[18], biomaterial deposition on to cylindrical mandrels[19,20], electrospinning[21], dip coating[22] and bioprinting[23]. However, few of them exhibit J-shaped stress–strain profiles and difficulties are reported in targeting vessel specific J-shaped curves or fine-tuning native vessel compliance. Some other strategies require maturation of cell-seeded grafts in specialized bioreactors[17,24], in vivo maturation and remodelling of synthetic grafts[25] or bicomponent configurations that account for the lag and linear region of the stress–strain J-shaped curve, resembling the biomechanical role of elastin and collagen fibres[26,27].

Cell seeding is required in many of the strategies above; however, this step is often ineffective in obtaining homogeneous cell distribution throughout the scaffold. On the other hand, acellularized options[25–27] lack biological functions. The benefit of cell-loaded grafts is exemplified where vascular constructs of natural/synthetic polymers cell seeded in pulsatile bioreactors have shown acceptable mechanical properties, improved cell infiltration and definitive tissue formation compared with their acellularized counterparts[24].

A strategy based on electrospinning and electrospraying of cells in culture media has proven feasible for the fabrication of vascular grafts with homogeneous cell distribution. However, it lacks control over the fabrication parameters to target specific J-shaped stress–strain curves and compliance values[28]. Nevertheless, none of these studies combine mimicry of the tri-layered structure of native arteries, the orientation of fibres in each layer and the distribution of different concentric cells patterned throughout the different layers, finely recapitulating mechanical properties (compliance) in a ready-to-use graft after rapid fabrication.

In this study, the J-shaped mechanical response and compliance of human coronary arteries were successfully reproduced by a bio-inspired strategy. This combines reinforced layers of cell-laden methacryloyl gelatin-alginate (GEAL) hydrogel with native-inspired arrangements of poly(ε-caprolactone) (PCL) sub-micron fibres in a fine-tuned, rapid and automated process. Additionally, the ready-to-use graft manufacturing permits control and customization of cell distributions in a concentric manner.

## Results

**Combined technology for vascular sublayer fabrication.** To obtain mechanically suitable grafts, multiple bilayers of tubular GEAL sublayer reinforced with PCL sublayers were fabricated, combining two different techniques: dip spinning[29] and solution blow spinning (SBS)[30], with the spinning apparatus positioned for fibre deposition at specified angles (Figs. 1 and 2a). Using a computer numerical-controlled (CNC) dip-spinning device, a cylindrical rod is moved vertically in synchrony to rotational movement. Fibres were sprayed towards the rod at specified angles by orientating the SBS head; simultaneous vertical movement of the rod (up-and-down) enabled the collection of aligned and oriented fibres, constituting a single cycle of PCL fibre deposition. A series of two fibre deposition cycles at opposite angle orientations (e.g. +/−21°) were applied to form a non-woven mesh-like PCL sublayer (Fig. 2a). The orientation of the fibres was adjusted by reversing the rotational movement of the rod, whilst maintaining the angle and position of the SBS head.

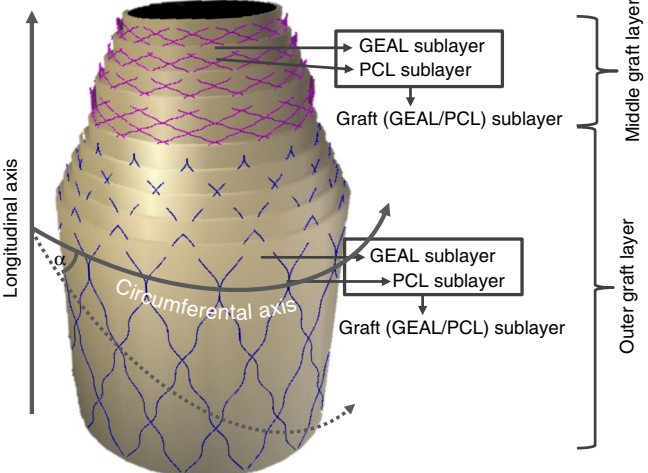

**Fig. 1** Scheme composition of the middle and outer graft layers. The middle graft layer comprises a series of four poly(ε-caprolactone)/methacryloyl gelatin-alginate (PCL/GEAL) sublayers, hereafter called middle graft sublayers, with fibres deposited at angles of +/−21° (+/− expresses the existence of a set of fibres at a defined angle X° respect to the circumferential axis, and a second set of fibres at an angle of 360 − X° with respect to the same axis) and GEAL sublayer generated after two cycles of dipping and photo-crosslinking. The outer graft layer is composed of a series of five PCL/GEAL bilayers, hereafter termed outer graft sublayers, with fibres deposited at angles of +/−67° and GEAL sublayer generated after three cycles of dipping and photo-crosslinking. The deposition angle is represented by α

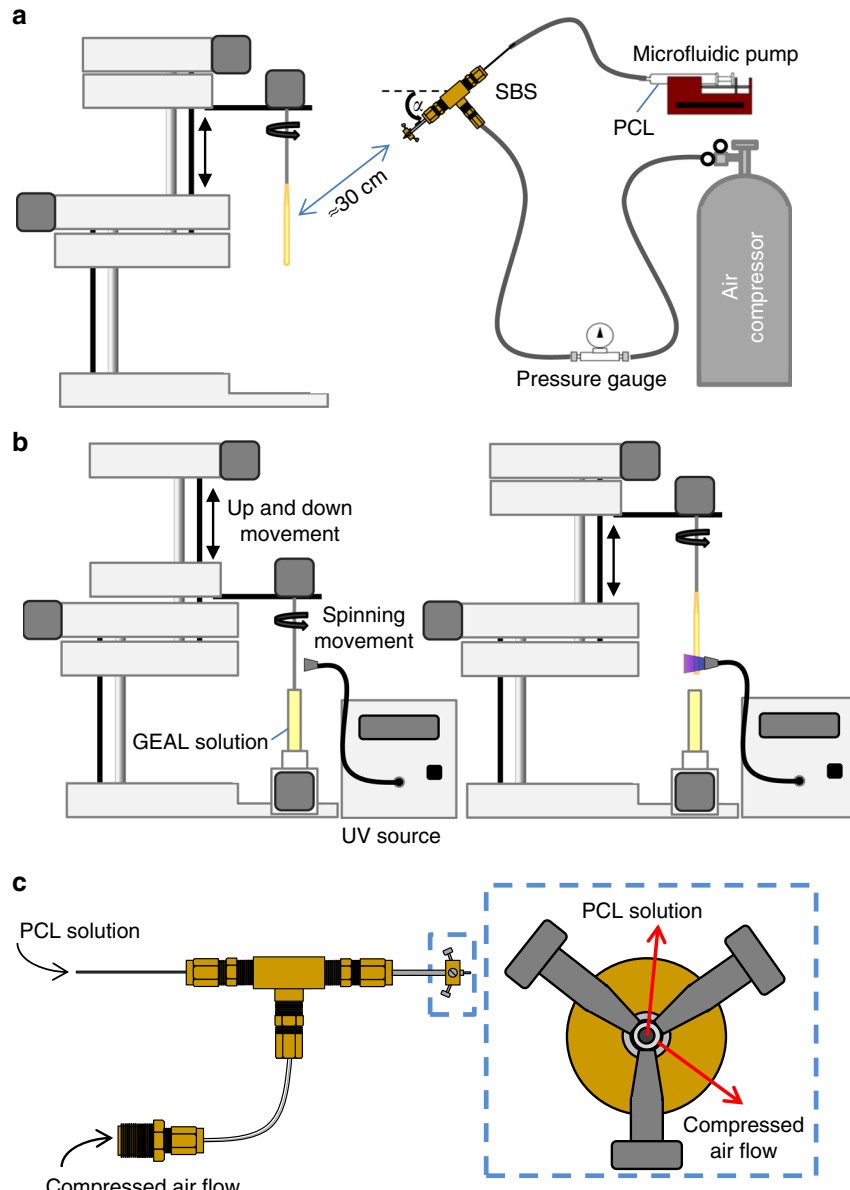

**Fig. 2** Fabrication method of small diameter vascular graft (SDVG) through the deposition of methacryloyl gelatin-alginate (GEAL) and poly($\varepsilon$-caprolactone) (PCL) fibres sublayers. **a** Fabrication of PCL fibre sublayer by deposition of PCL fibres into a spinning rod via angled solution blow spinning. During this process the rod moves vertically via the dip-spinning system. **b** Fabrication of GEAL sublayers by vertical dipping of an orientated fibre-coated rod into GEAL solution followed by rotational retraction and exposure to ultraviolet (UV) light. **c** Solution blow-spinning (SBS) apparatus showing a central-inner nozzle for the extrusion of PCL solution, and a concentric-outer nozzle for delivery of compressed air (right, end view). The central-inner nozzle protrudes 1 mm from the air nozzle

Subsequently, the rod containing the orientated fibres was immersed into the GEAL solution and slowly retracted whilst spinning to allow homogeneous GEAL layer photo-crosslinking using a lateral ultraviolet (UV) source (Fig. 2b). Several cycles of dip spinning/crosslinking enable a stable concentric layer of GEAL on top of the fibres. This process was repeated to build up multiple hydrogel layers intercalated with PCL fibre layers deposited at the desired angles.

To reproduce the arrangement of collagen fibres observed in blood vessels, PCL sublayers comprising two intercalated sets of fibres at opposite orientations were fabricated. To understand the extent to which fibre angles govern the mechanical behaviour of the multilayer construct, the SBS apparatus was adjusted either at angles of 21° (Fig. 3a) or 67° (Fig. 3d). Fibre deposition angle is defined by the angular orientation of the SBS head with respect to

the circumferential axis (see Fig. 1 and Supplementary Fig. 1). PCL fibre sublayers fabricated with a target of +/−21°, and had resultant average fibre angles of 31 ± 31° (Fig. 3b) in one orientation and −28 ± 32° (Fig. 3c) in the opposite orientation, with an average fibre diameter of 700 ± 250 nm. The PCL fibre sublayer fabricated with a target of +/−67°, and exhibited fibre angles of 78 ± 22° (Fig. 3e) and −77 ± 22° (Fig. 3f), with an average fibre diameter of 1.2 ± 0.5 µm. Smaller diameters may be due to fibre stretching caused by rod rotation and should have a greater effect on fibres targeting shollow angles (e.g. +/−21°). Stiffer mechanical responses were observed for PCL fibre sublayers fabricated at +/−67° angles compared to +/−21° as evidenced from the stress–strain profiles of longitudinal tensile tests (Fig. 3g, h), due to the higher degree of fibre alignment in the testing direction and fibre diameter. The opposite is expected for

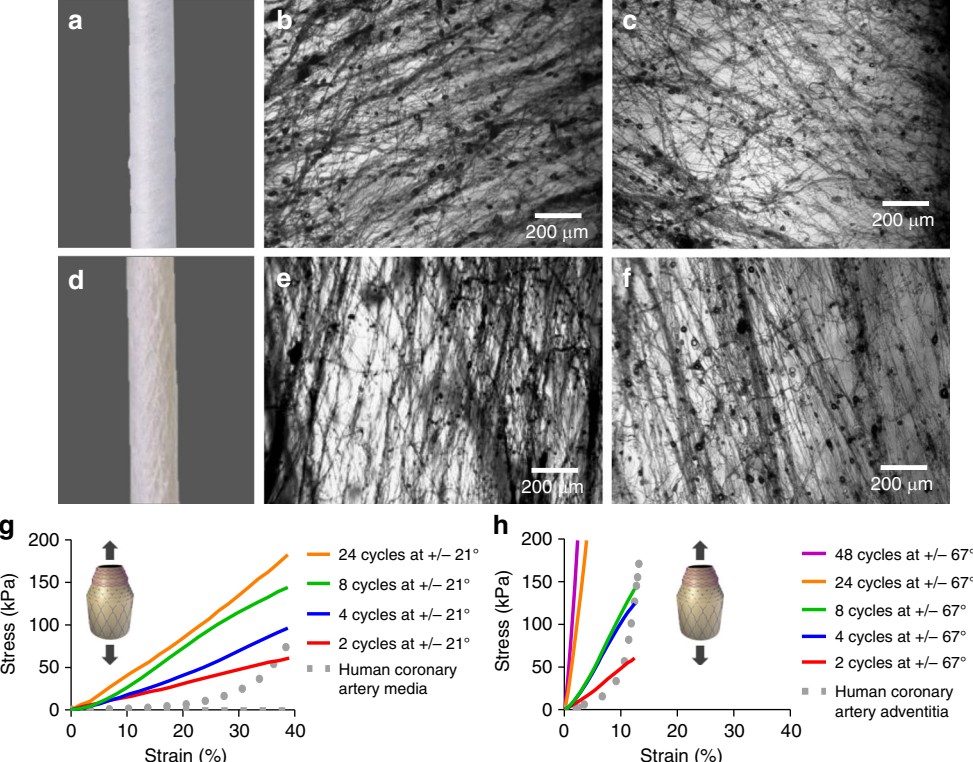

**Fig. 3** Poly($\varepsilon$-caprolactone) (PCL) sublayers. **a** Macroscopic inspection of PCL fibres at 21° angle co-deposited in both orientations (+/−21°). **b, c, e, f** Optical microscopy of fibres obtained after one cycle of PCL fibre deposition using individual orientations. **b, c** Single cycle of PCL fibre deposition at +/−21° angle using a clockwise (+21°) and an anti-clockwise (−21°) rod rotation, respectively. **d** Mascroscopic inspection of PCL fibres at 67° angle co-deposited in both orientations (+/−67°). **e, f** Single cycle of PCL fibre deposition at +/−67° angle using a clockwise (+67°) and anti-clockwise (+/−67°) rod rotation, respectively. **g, h** Stress–strain curves for longitudinal tensile testing conducted on PCL sublayers fabricated at deposition angles of +/−21° and +/−67°, respectively. PCL sublayers were manufactured using different number of cycles of intercalated PCL fibres deposited at opposite angles (+/−). Every fibre deposition cycle using a clockwise rotational direction is alternated with a second fibre deposition cycle using an anti-clockwise rotational direction (+/−); therefore, two cycles constitute one PCL fibre sublayer (see Fig. 1). Dotted lines correspond to average stress–strain curves for longitudinal tensile tested human coronary media layers (**g**) and human coronary adventitia layer (**h**). Human coronary control data were collected from previous studies[11]. Three experiments were conducted for each condition and the averaged curves displayed in the plot

circumferential tensile testing, where the testing direction has a higher alignment with fibres deposited at +/−21° (see Fig. 1). The J-shaped mechanical behaviour of native arterial vessels was not attained even after adjusting the number of PCL sublayers (see Fig. 3g, h), and therefore, two novel approaches were developed to improve J-shaped matching, described below.

**Pre-stretching and wavy PCL fibre deposition.** To match the J-shaped stress–strain curves[13], 24 PCL fibre sublayers were deposited at +/−21° (48 individual layers) and preconditioned after fabrication by stretching cycles before subjecting the construct to longitudinal tensile testing. PCL sublayers were subjected to a preconditioning step consisting of five loading/unloading cycles, applying different percentages of strain at a constant rate of 10 mm min$^{-1}$. Through iterative improvement, the strain during preconditioning was set to 30% (Supplementary Fig. 2). This generated a sigmoidal curve, approaching the natural J shape (see Fig. 4e). Efforts were made to mimic the wavy collagen and elastin fibre morphologies observed in native vessels[12,31,32]. Waviness was introduced by rapid alternation of clockwise and anti-clockwise rod rotations during fibre deposition (Supplementary Fig. 3). By using an unbalanced ratio of clockwise and anti-clockwise rod rotation (e.g. 1 s in one direction and 0.5 s in the opposite direction), the system enabled wavy PCL nanofiber deposition (Fig. 4b, d). Preconditioning and unbalanced

clockwise/anti-clockwise rotation resulted in closer mimicry of the J-shaped stress–strain curve of native vessels[12,31,32] (see Fig. 4e).

**Mechanical improvement of the middle and outer graft layers.** To define the structural configuration of the middle and outer graft layers of the new bio-inspired small diameter vascular graft (SDVG), iterative testing of layered constructs were performed to match the stress–strain profiles of the media and adventitia layers of human coronary arteries[11]. Each bilayered sublayer (graft sublayer) was composed of one wavy PCL sublayer (e.g. +/− 21°) and one GEAL sublayer (see Fig. 1). Inspired by the distribution of collagen/elastin fibres and cells in human arteries, this inter-calated configuration of PCL fibre sublayers and cell-laden GEAL sublayers allowed the concentric patterning of cells. The number of graft sublayers (PCL/GEAL sublayer) comprising the middle and outer layers were adjusted to fit mechanical properties of the media and adventitia layer. The GEAL sublayers constitute an important element in defining the mechanical behaviour, and therefore, the middle and outer graft layers were specifically formulated to achieve close mechanical fitting with the media and adventitia layers. Each GEAL sublayer of the middle graft layer was generated after two cycles of dip spinning in GEAL solution and crosslinking, whereas GEAL sublayers for the outer graft layer were generated after three cycles of dip spinning/cross-linking. Stress–strain curves of the middle and outer layers were

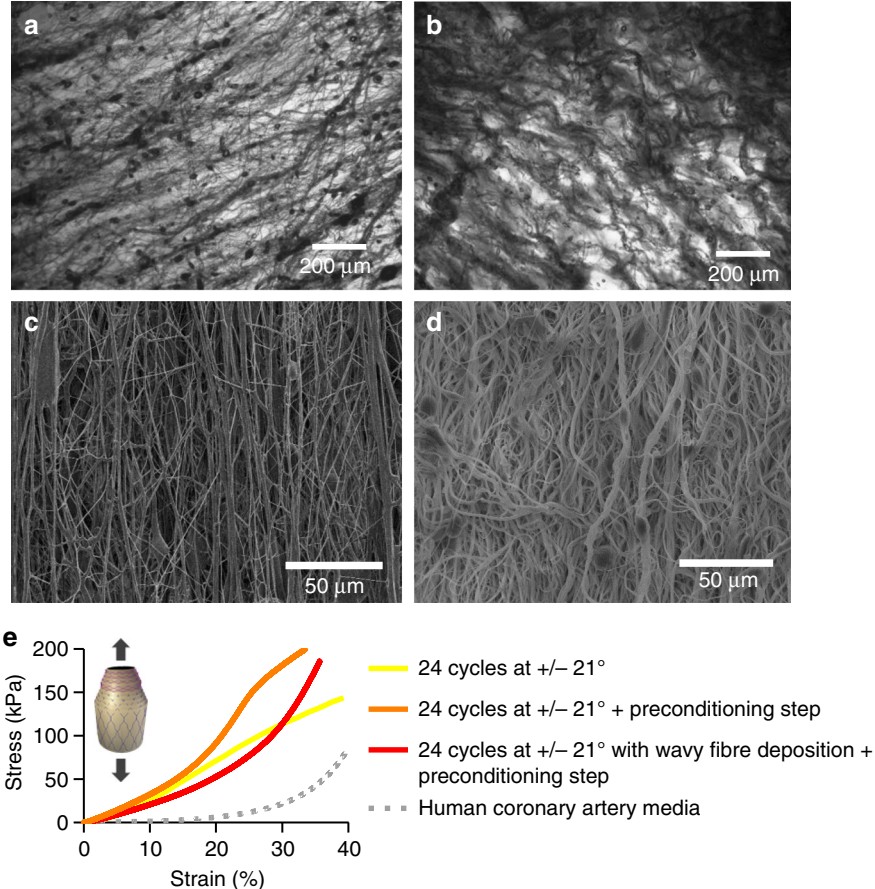

**Fig. 4** Towards a J-shaped stress–strain curve combining wavy fibre deposition and preconditioning: **a** Optical microscope image of a poly($\varepsilon$-caprolactone) (PCL) fibre sublayer fabricated at a deposition angle of 21° after one cycle of fibre deposition with continuous clockwise rod spinning. **b** Optical microscope image of a PCL fibre sublayer fabricated at a deposition angle of 21° after one cycle of fibre deposition with alternated rod spinning and after the preconditioning step. **c** Scanning electron microscopy (SEM) image of a series of 24 PCL fibre sublayers fabricated at a deposition angle of +/−67°. **d** SEM image of a series of 24 PCL fibre sublayers fabricated at a deposition angle of +/−67° with alternated rod spinning and after the preconditioning step. **e** Longitudinal stress–strain curve for a series of 24 PCL fibre sublayers fabricated with deposition angles of +/−21°, with and without a stretch preconditioning step of five cycles of loading/unloading at 30% strain and wavy fibre deposition using the alternating rod spinning during angled fibre deposition. The grey line represents previously published stress–strain mechanical behaviour of the media layer of the human coronary artery under longitudinal tensile testing[11]. One independent experiment was conducted for each condition

plotted as a function of numbers of middle and outer sublayers (Fig. 5a). The outer graft layer fabricated with five outer graft sublayers containing fibres at +/−67° exhibits a stress–strain curve resembling the adventitia layer, whereas the middle graft layer fabricated with four middle graft sublayers and fibres at +/−21° resembles the media layer (Fig. 5a).

The middle and outer graft layers had a thicknesses of 323 ± 41 and 298 ± 42 μm, respectively, which together produces a wall thickness similar to the human coronary arteries (750 ± 170 μm)[33]. Each layer exhibited anisotropic non-linear mechanical response similar to the media and adventitia of human coronary arteries. The outer graft layer exhibits stiffer behaviour in the longitudinal direction compared to the circumferential direction (Fig. 5b, c), whereas the middle graft layer exibits the contrary, as expected (Fig. 5d, e).

To visualize the intercalated distribution of the outer and middle layer after the combined fabrication using the dip-spinning CNC and SBS, a full SDVG was fabricated, mimicking the three layers of native arteries: the inner (intima), middle (media) and outer (adventitia) graft layers. The inner graft layer comprised nine cycles of dip spinning/crosslinking in GEAL solution; middle and the outer layer were fabricated as described

above, constituting four middle graft sublayers and five outer graft sublayers (PCL/GEAL sublayers). Complete SDVG manufacture took 30 min once the precursor solutions were prepared. The wall thickness of the resultant SDVG was 0.59 ± 0.17 mm with an inner diameter 3.6 ± 0.5 mm.

The nine graft sublayer structure of the SDVG was investigated by X-ray computer tomography (CT) (Supplementary Fig. 4 using a freeze-dried sample). All nine graft sublayers are distinguished in the transversal cut and micro-CT visualization. The transverse section shows the PCL fibre sublayers separated by the GEAL hydrogel sublayers (apparent void space) (Supplementary Fig. 4b). Fibres appear continuous and individualized, with minimal fibre fusion evident, according to scanning electron microscopy (SEM) and CT (Fig. 4d and Supplementary Fig. 4c, respectively). Fibre fusion is of concern in the design of the SDVG, as this would impact the mechanical response, mainly due to force distributions at fibre contact points. Overall in the SDVG, free displacement of fibres would be possible during stretching and recoil. Porosity was evaluated using CT data for an SDVG without including GEAL sublayers, and was determined as 68 ± 11% by post-processing image analysis (Avizo), taking five regions in the construct (Supplementary Fig. 4d).

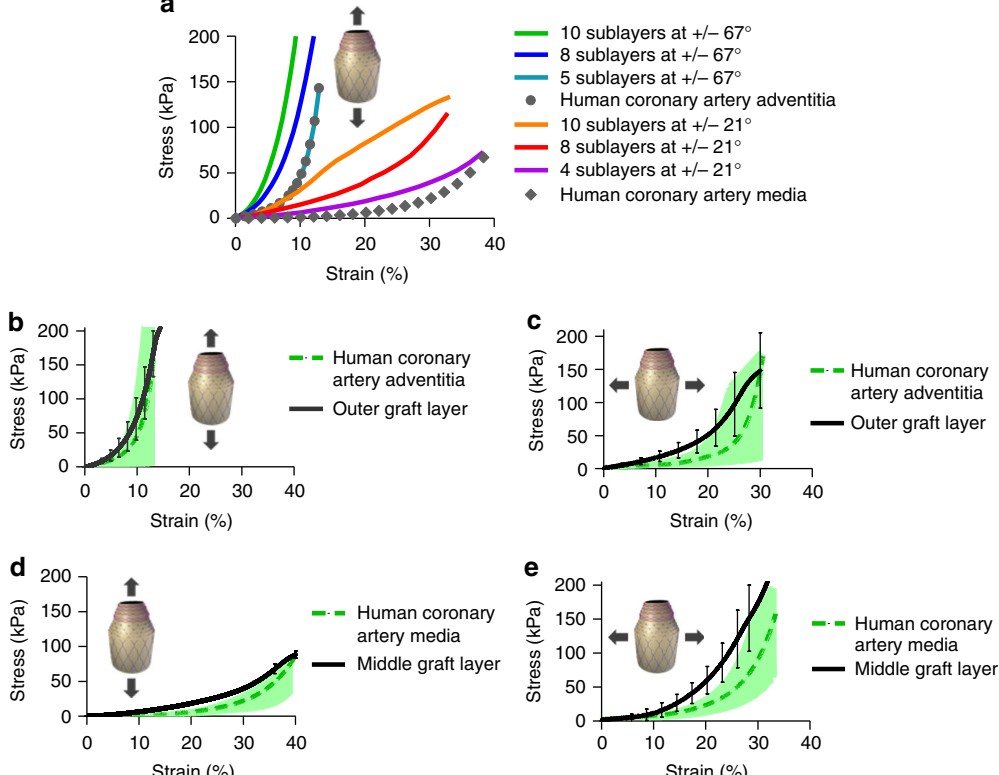

**Fig. 5** Stress–strain curves of the outer and middle graft layers based on methacryloyl gelatin-alginate (GEAL) reinforced poly($\varepsilon$-caprolactone) (PCL) sublayers. **a** Iterative improvement of the middle and outer graft layers: longitudinal tensile testing of the middle and outer graft layers using different numbers of middle and outer graft sublayers in the construct. Grey dotted lines represent the average longitudinal stress–strain curve of the native adventitia (closed circles) and media (closed diamonds) layer of human coronary arteries[11]. **b**, **c** Longitudinal and circumferential stress–strain curve of the outer graft layer composed of five graft (GEAL/PCL) sublayers. Green dashed lines in **b**, **c** represent the average longitudinal and circumferential stress–strain curves of the native media layer of coronary arteries[11], respectively. **d**, **e** Longitudinal and circumferential stress-strain curve of the middle graft layer composed of four graft (GEAL/PCL) sublayers. The green dashed lines in **d**, **e** represent the average longitudinal and circumferential stress–strain curve of the native adventitia layer of human coronary arteries[11], respectively ($n = 3$). The shaded green zones in the figures represent the range of results obtained for native human coronary arteries[11]. Error bars = standard deviation; $n = 3$ independent experiments

**Tensile testing, compliance, burst pressure and suture retention**. Non-linear and anisotropic mechanical response is maintained when tested together in the configuration of a full SDVG, comprising of inner, middle and outer graft layers, GEAL sublayers, with wavy, orientated PCL fibre sublayers and pre-conditioning. The construct exhibited mechanical response similar to human coronary arteries, both in longitudinal and circumferential directions (Fig. 6a, b). Longitudinally tested SDVG exhibited a maximum failure strength of $520 \pm 56$ kPa, which is in the range of coronary arteries for individuals above 35 years of age[13,34] (Supplementary Fig. 5). To verify whether the mechanical properties of the SDVG change upon repeated loading and unloading cycles, a fatigue assay consisting of 20 repetitions of circumferential strain up to 25% was performed (Fig. 6c). A linear response is shown in the first cycle (preconditioning step), after which non-linear anisotropic behaviour is observed, converging to the desired J-shaped curve, with minimal hysteresis over 20 cycles, similar to previous studies conducted in human coronary arteries[11].

Vascular constructs were submitted to pressurized testing using a perfusion system with pulsatile luminal pressure changes. To test the pressurizing behaviour at physiological axial strain, constructs were fixed to 10% axial stretching (longitudinal) prior to testing. Vascular grafts may be submitted to larger longitudinal stretching after the surgical graft implantation; therefore, two additional stretching degrees of 20% and 25% axial strain were

applied and fixed prior to pressure testing[35]. Comparisons with published compliance values of human coronary arteries[34,35] were conducted to evaluate the similarity of the SDVG to natural vessels. Small progressive increments in the diameter change ratio ($D/D_0$) were obtained during pressure variation (Fig. 6d–f). Longitudinal and circumferential preconditioning was conducted after installation of the graft in a perfusion system prior to fixing the stretch level and pressure testing (see Methods). SDVGs showed an increase in external diameter under luminal pressurization, similar to human coronary arteries, at the in vivo axial strain value for blood vessels (10–20%)[35,36] (Fig. 6d, e). At 25% axial stretching, larger changes of nominal diameter were observed compared to human coronary arteries (Fig. 6f).

Calculated compliance values for SDVGs at 10% and 20% axial stretching during pressure testing showed no statistical difference with reported results for human coronary arteries (Supplementary Table 1) in the physiologic pressure range (80–120 mmHg). These results demonstrate that SDVGs have a coronary compliant response. At higher values of axial stretching (25%), compliance values of human coronary arteries and SDVGs were statistically different (Supplementary Table 1), hence the level of axial stretching after bypass surgery would need to be considered for mechanical matching with the anastomosed native vessel. Interestingly, compliance of SDVGs at lower pressure range (50–90 mmHg) was higher in comparison to human coronary arteries of older patients, yet closer values to younger individuals[28].

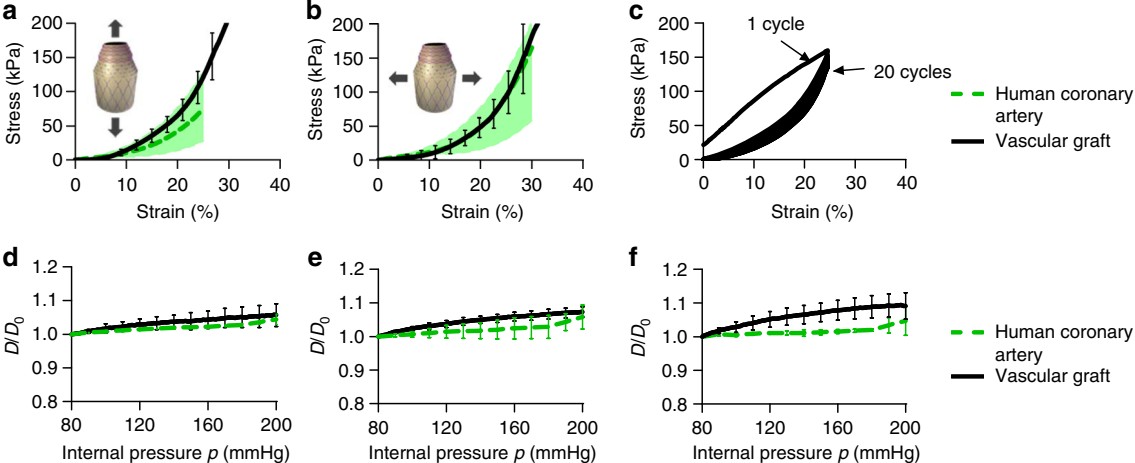

**Fig. 6** Mechanical evaluation of the fabricated small diameter vascular graft (SDVG). **a**, **b** Stress–strain curves of the SDVG (black line) and the human coronary artery in **a** longitudinal and **b** circumferential stretching directions ($n = 5$ independent experiments). The green dashed lines and green shaded zones in **a** and **b** represent average and value range, respectively[35]. The green shaded zones represent the range of results obtained for native human coronary arteries[35]. **c** Cyclic tensile testing in the circumferential direction. **d–f** Profiles of diameter change ratio ($D/D_0$) as function of pressure applied to the SDVG (black line, $n = 5$ independent experiments) compared with human coronary arteries (green dashed lines, $n = 5$ independent experiments)[34,35] at three different values of axial pre-stretch during testing ($e_z$). **d** $e_z = 10\%$ of axial pre-stretch. **e** $e_z = 20\%$ of axial pre-stretch. **f** $e_z = 25\%$ of axial pre-stretch. Error bars = standard deviation

Due to safety concerns, vascular grafts must have adequate burst pressures and suture retention strength[20,36]. Full SDVGs exhibited burst pressures of $1630 \pm 180$ mmHg, similar to values reported for human saphenous veins, used in coronary bypass[37] (Supplementary Fig. 6). Suture retention strength was $143 \pm 13$ grams-force, similar to reported values for human internal mammary arteries[37] (Supplementary Fig. 7).

**Cell distribution and viability in cell-laden SDVGs.** Human umbilical cord endothelial cells (HUVECs) were added to GEAL solution to a concentration of $10^7$ cells mL$^{-1}$ prior to manufacturing. This concentration was previously established as adequate for concentric cell-laden hydrogel formation[29]. A 14-μm-width transverse cryo-section of the SDVG was nuclear stained to quantify the encapsulated cells and verify the cell distribution throughout the wall thickness (Fig. 7d, e). A higher cell number per unit volume is observed compared to the cell number in the GEAL solution ($22 \times 10^4$ vs. $1 \times 10^4$ cells mm$^{-3}$). This cell accumulation can be attributed to gravitational draining of the viscous solution and concomitant intercapillary forces between cells during rod withdrawal after dipping[38]. SDVGs exhibited homogeneous and concentric cell distribution (Fig. 7e). Using bone marrow-derived mesenchymal stem cells (BM-MSCs), and a LIVE/DEAD® cell staining assay, cell viability analysis within different sublayers of the SDVG was performed at different time points during static cell culture (Fig. 7f–j). Evaluation on day 7, 14, 21 and 28 showed 71%, 84%, 87% and 92% viability, all respectively (Supplementary Fig. 8). These results confirm cell survival and proliferative capacity post fabrication (Fig. 7k).

**Engraftment potential of a cell-laden bio-inspired SDVGs.** Considering the incorporation of a cell essential for the design and successful outcome in a transplantation scenario[24,39], important implication must be taken into account when choosing appropriate cell types. Although autologous vascular cells are the preferable choice, invasive harvesting and long-term culture make this a risky option in terms of clinical and commercial feasibility. Induced pluripotent stem cells are potentially an excellent source[40], due to their autologous nature and low invasiveness of

harvesting; however, reprogramming and differentiation remain long and expensive procedures, and the frequency of point mutations[41] has generated serious safety concerns. Allogenic BM-MSCs require less invasive harvest, are storable, immuno-tolerated, ecocomically feasible and have been implicated in vascular repair and remodelling[42]. Additionally, BM-MSCs are known to have immunomodulatory activity; therefore, inflammation in the presence of this cell type could be controlled or ameliorated. BM-MSCs were therefore selected to add biological functionality in the SDVG design. Analysis of immunocompetent mice with dorsal subcutaneous implantations demonstrates that SDVGs with encapsulated BM-MSCs can control an inflammatory response, whereas non-cellularized SDVG did not (Supplementary Fig. 9, 10 and Supplementary Note 1 for further details). This further demonstrates that cells maintain viability and functionality after being subjected to SDVG manufacture. In this study, vascular cell functionality, such as contractility, was not under evaluation, instead functionality of BM-MSCs[43].

Platelet activation plays a central role in coagulation and thrombus stabilization, a process initiated after platelet adhesion to exposed collagen, activation and binding of fibrinogen to platelet membrane receptors[44]. A flow cytometry-based platelet activation assay was used to evaluate exposure of human platelet-rich plasmas (PRPs) to the fabricated SDVGs with and without BM-MSCs. This assay was conducted using individual elements of SDVG fabrication (BM-MSCs, crosslinked GEAL and encapsulated BM-MSCs). To quantify the level of platelet activation, exposed PRPs were immunolabelled: anti-CD42a to distinguish platelets, and anti-CD63 and anti-CD62P to evaluate the activation degree. PRPs exposed to SDVGs in the presence or absence of BM-MSCs resulted in CD62P$^+$ and CD63$^+$ platelet populations with lower frequency than values considered as activated[45] (see Fig. 8a, b). Similarly, low levels of CD62P$^+$ and CD63$^+$ platelet populations were obtained with PRPs exposed to the individual elements of the SDVG (see Fig. 8a, b). Only when evaluating CD62P$^+$ platelets did acellular SDVG appear significantly higher than SDVG with BM-MSCs (Fig. 8a).

To unveil the potential clotting induction of SDVGs due to blood–graft interface contact activation, human whole blood was subjected to luminal graft contact and incubated for different

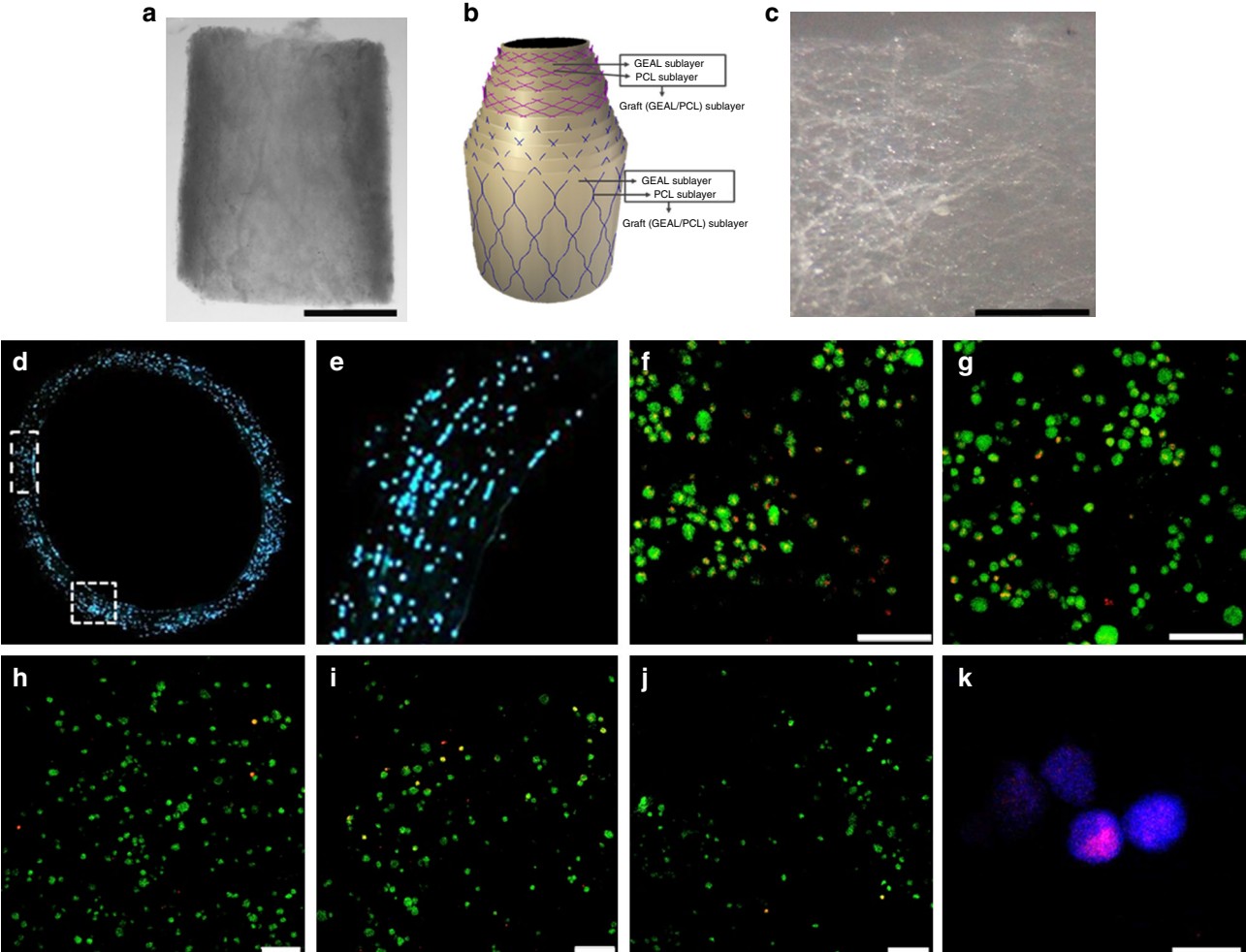

**Fig. 7** Density and distribution of cells within the methacryloyl gelatin-alginate (GEAL) sublayers of the fabricated small diameter vascular grafts (SDVGs). **a** Macroscopic image of a segment of the SDVG (scale bar = 2 mm). **b** Schematic representation of the SDVG multilayer configuration (see also Fig. 1). **c** A single isolated graft sublayer from a fabricated cellularized SDVG (scale bar = 200 μm). This particular imaged layer corresponds to a middle graft sublayer (PCL/GEAL sublayer). **d** Fluorescence microscopy image of a transversal section of an SDVG with cell nucleus staining (Hoechst 33342). **e** Higher magnification image of the vessel wall in a transversal cut, showing cells present throughout the thickness of the SDVG, from the lumen to the outer layer (right to left). **f** Confocal image reconstitution of 100 μm z-stacked images showing LIVE/DEAD® cell staining in the SDVG after 2 weeks of static cell culturing (viable cells = green fluorescence; dead cells = red fluorescence) (scale bar = 100 μm). **g** LIVE/DEAD® cell staining after 3 weeks of static cell culturing (confocal image reconstitution of 100 μm z-stacked images). **h–j** Representative confocal images of LIVE/DEAD® cell staining of the SDVG after 4 weeks of static cell culturing (scale bar = 100 μm). From the lumen to the outer layer, **h** is the second, **i** the sixth and **j** the eighth individually isolated graft sublayer. **k** Ki-67 nuclear staining in cells embedded in one graft sublayer demonstrating proliferative state (scale bar = 50 μm). Ki-67 staining is only present in active phases of the cell cycle (G1, S, G2 and mitosis), and absent in quiescent cells

time periods. A commercial ePTFE graft, although not indicated for small bore vessels, was included as gold standard (GORE-TEX® Vascular Graft, #RRT080700). Comparative evaluation of acellular SDVGs exhibited similar clotting profiles to the commercial vascular graft (Fig. 8c); however, cellularized SDVGs showed significantly increased blood clotting at 5 and 10 min of incubation compared to ePTFE and acellularized SDVG. This is possibly related to cell-derived tissue factor secretion, which has been previously reported for BM-MSCs[46].

A preliminary study in a rabbit carotid graft model was conducted to assess the implantability of SDVGs; the internal diameter of SDVGs was adapted to match the internal diameter of rabbit carotid arteries (RCA) (1.5 mm). Since the SDVG was designed to dimensionally and mechanically match the human coronary artery, neither the wall thickness nor the mechanical behaviour of the SDVG exactly matched those of RCAs. Two rabbits were implanted with acellular SDVGs and two with

BM-MSC cellularized SDVGs, and compared to two carotid incision-anastomosis controls. After surgery, no blood leakage was observed for any of the experimental group, demonstrating appropriate suturing and quick haemostasis. Proximal blood flow was evidenced after 12 h approximately post surgery for all implanted SDVGs (Fig. 8g). Although patency was only initially observed in the in vivo experiment, implanted SDVGs in one of the two animals per group was extracted 14 days post surgery (Fig. 8d, h), and the others at 30 days for histological evaluation (Fig. 8i). As expected, thrombus formation was identified in the luminal section of the implanted SDVG (Supplementary Fig. 11). Comparing haematoxylin and eosin stain (H&E) of the BM-MSC-laden SDVG extracted on days 14 and 30, partial and complete cellular invasion/remodelling of the grafts was observed, respectively (see Fig. 8h, i and Supplementary Fig. 11b). The acellularized SDVG at day 30 revealed limited remodelling in the sections proximal to the lumen (Supplementary Fig. 11a).

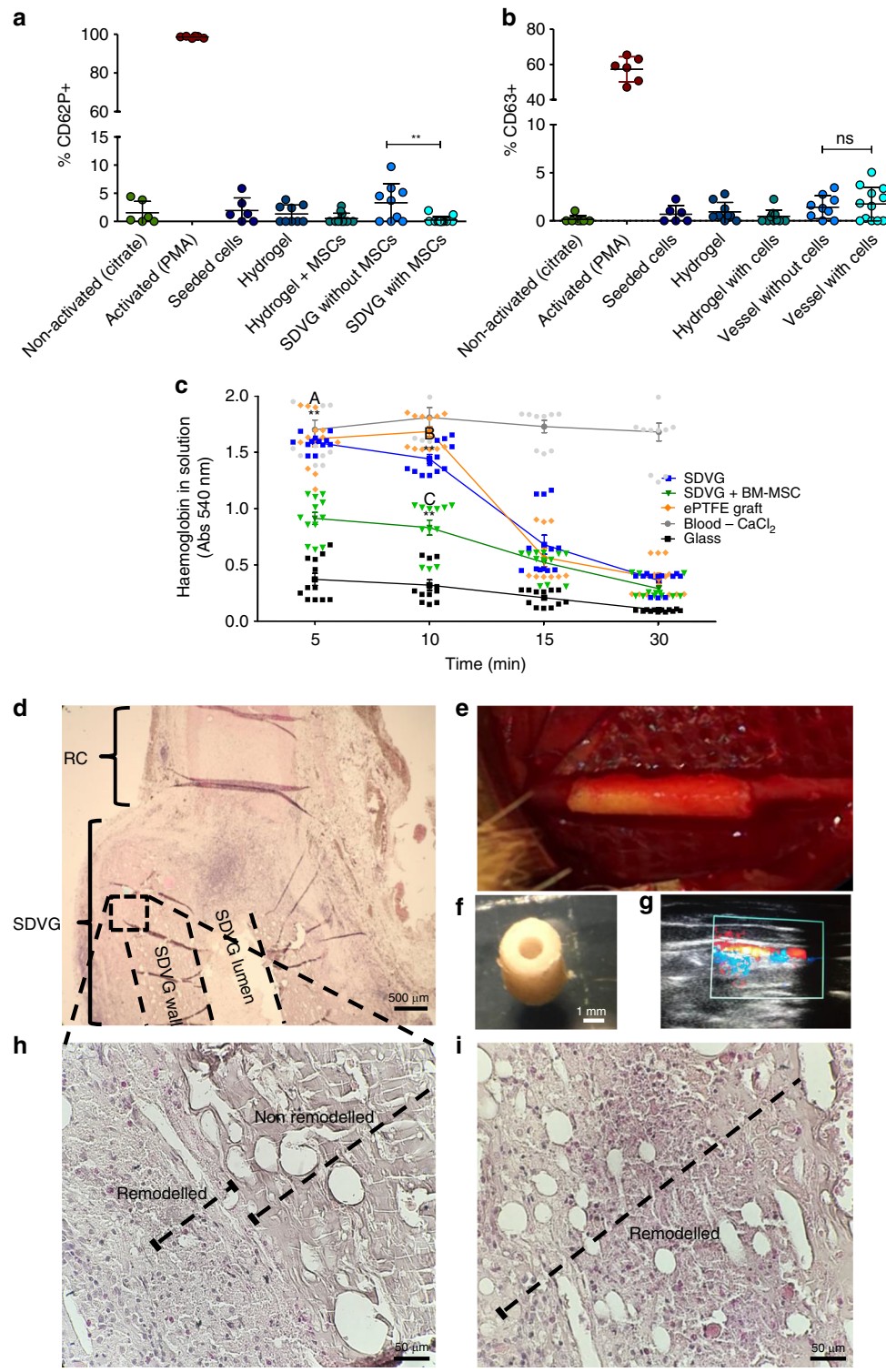

## Discussion

In this study, vascular grafts resembling the mechanical behaviour of human coronary arteries were successfully fabricated by combining the dip-spinning method[29] and adapted SBS device for angled fibre spinning that also enabled fibre waviness to be imparted. This manufacturing method allows the reinforcement of the cellularized GEAL layers with PCL fibres, intercalating GEAL and PCL fibre sublayers. The compositions of these grafts were iteratively improved to reflect the mechanical properties and behaviour of a human coronary artery. This method can be easily customized to mimic other native blood vessels; the mechanical properties can be customized by changing the number of dippings, the quantity of PCL fibre deposited during fabrication and the orientations of the fibres. Furthermore, different cell types can be encapsulated in the GEAL sublayers to impart certain biological activities and assist the in vivo remodelling of grafts.

Vascular grafts have previously been fabricated for the replacement of small diameter blood vessels[16,18–20]. However, most studies into tissue-engineered vascular grafts as replacement coronary arteries have failed to attain the native J-shaped

**Fig. 8** In vitro and in vivo experimental evidences of implantability potential. **a**, **b** Platelet activation of human platelet-rich plasma (PRP) exposed to small diameter vascular graft (SDVG). All cell events considered in the flow cytometer analysis are CD42a[+] (platelets). **a** CD62P[+] cell events and **b** CD63[+] cell events obtained from immunostained PRP samples exposed to SDVG and individual elements of the SDVG. PMA = 4β-Phorbol 12-myristate 13-acetate. Error bars = standard error of the mean. **p < 0.01; ns: non-significant. $n = 3$ PRP donors with experiment triplicates. Statistical analysis was conducted using the Mann–Whitney $U$ test. **c** Clotting time assay. Whole human blood samples were incubated in direct contact with surfaces of different materials and vascular grafts, including the commercial GORE-TEX® Vascular Graft, and acellularized and cellularized bio-inspired SDVGs. Red blood cells not contained into a blood clot are subjected to lysis and the amount of haemoglobin quantified through absorbance at 540 nm. Error bars = standard error of the mean. $n = 4$ blood donors with experimental triplicates. "A" indicates significant difference between acellularized SDVG and bone marrow-derived mesenchymal stem cell (BM-MSC)-laden SDVG after 5 min of incubation, "B" between expanded polytetrafluoroethylene (ePTFE) graft and acellularized SDVG after 10 min of incubation and "C" between acellularized SDVG and BM-MSC-laden SDVG after 10 min of incubation. **p < 0.01. Statistical analysis was conducted using the Mann–Whitney $U$ test. **d–i** Rabbit carotid grafting model. Preliminary evaluation of grafting feasibility in an artery circulation system. **d** Haematoxylin and eosin (H&E) stain of a saggital cut at the anastomosis section (×4). **e** Anastomized SDVG on day 0 at the end of the surgical procedure. **f** Stereomicroscopy transversal image of the SDVG with the adapted luminal dimension for rabbit carotid grafting. **g** Evidence of eco-Doppler blood flow at the carotid section proximal to the SDVG anastomosis. **h** Microscopy image (×40) of a H&E-stained saggital cut of the SDVG wall after 14 days of arterial implantation and **i** at after 30 days

mechanical response[37]. To overcome these issues, the middle and outer layers were fabricated to mimic the structural configurations of collagen and elastin fibres in the media and adventitia layers of the coronary artery, respectively. The elastin and collagen fibres are known to exert their action over different pressure ranges; at low pressures artery deformation is dominated by the elastin, whereas at high pressures, collagen fibres start to align and govern the mechanical response[48]. Inspired by this structure, reinforcement fibres were deposited using a rod rotation sequence, thereby the fibres are not fully stretched during the fabrication, and exhibit a wavy pattern, similar to collagen fibres in natural vessels in unloaded pressure state[12,31,32]. Blood vessels are in constant loading–unloading cycles, triggering cells and extracellular matrix adaptations that contribute to the J-shaped mechanical behaviour[49]. In the fabrication of SDVGs, a preconditioning step was applied ahead of tensile and pressurization tests, and demonstrated to be essential in attaining the native mechanics of human coronary arteries. As detailed in Methods section, preconditioning of full SDVG comprises a series of five cycles of 30% longitudinal strain and five cycles of luminal pressure loading and unloading up to 200 mmHg, thus the fibres can re-adapt and align within the scaffold. It may even be possible to conduct the preconditioning in situ on implantation by pulsatile blood pressure without the need of previous preconditioning, which is the subject of future studies.

SDVGs based on GEAL reinforced with PCL fibres were fabricated combining the inner and individually optimized middle and outer layers. The existence of an inner layer has the perspective to control coagulation either after endothelization in vivo or through the encapsulation of anti-coagulant factors; therefore, although not tackled in the present study, there is room for improvement in the anti-coagulant properties of the inner luminal layer. The stress–strain curves shown for the SDVG fabricated using the combination of the three layers fits within the range of human coronary arteries in both circumferential and longitudinal tensile testing (Fig. 6). The inner diameter and thickness are also comparable with the human coronary arteries, which have been reported to range from 3.7 to 1.9 mm[50] and 0.55 to 1 mm[51], both respectively.

The low patency of previously developed SDVGs has been associated with intima hyperplasia and thrombus formation[8], mainly related to luminal diameter and mechanical mismatch with the native vessel[52,53]. These are causes of flow vector changes and tissue stretching at the anastomosis zone, generating damage to the tight endothelial layer and neointimal hyperplasia, respectively[15]. The replacement of blood vessels with a SDVG that has luminal diameter and mechanical behaviour identical to native vessels will reduce the possibility of intimal hyperplasia formation by decreasing the flow and tension variations at the anastomosis zone. This new manufacturing method produced SDVGs that exhibit similar diameter variation in response to physiological blood pressure. SDVG compliance is comparable to human coronary arteries at physiological pressures and stretch (Fig. 6 and Supplementary Table 1). This demonstrates the effectiveness of optimizing each layer separately to achieve the production of SDVGs with suitable mechanical properties, and highlights the importance of the effect of fibre angle deposition, waviness and preconditioning on mechanical properties.

The combined techniques used in this work allow the controlled and homogeneous incorporation of cells throughout the SDVG in the GEAL sublayers, which is otherwise difficult to achieve via post-fabrication cell seeding. Cells remained viable during fabrication, and demonstrated proliferative and functional capacity within the SDVG (Fig. 7 and Supplementary Figs. 8, 9 and 10). In vivo immunogenicity experiments (Supplementary Fig. 10) demonstrate that the encapsulation of BM-MSCs lowers the inflammation response caused by implantation. Immunosuppressive cell function is of importance in the design of a new vascular graft, considering that a cause of graft failure is associated to chronic inflammatory response[4]. These results prove the importance of fabricating cellularized grafts with BM-MSCs to prevent inflammation and enhance implant functionality by promoting cell repopulation and definitive tissue remodelling after implantation[54]. The PCL fibres present in the SDVGs have been selected to degrade within a year (tunable)[55], while the GEAL component can be remodelled and degraded within a few months[56]. This biomaterial combination enables continuous tissue regeneration through remodelling of the hydrogel while maintaining mechanical properties that could help to prevent aneurysmal-like dilation. BM-MSCs can be encapsulated as a biological component with the ability to assist in the recruitment and migration of the patient's own cells by secreting various chemokines, and help in the formation of a definitive antithrombogenic inner layer after implantation[54].

SBS has significant advantages over other manufacturing methods commonly used for the fabrication of fibrous scaffold, such as electrospinning, melt electrospinning or direct writing[47,57–59]. SBS is applicable over a wide range of materials, including sol-gel hybrids[60], bioactive composites[61] and natural materials such as silk[62]. In this work, SBS has been adapted to enable the facile deposition of fibres at relatively defined angles, to create non-woven orientated veil-like fibre structures at high fibre deposition rates. The technique does not require high voltage-derived electric fields, enabling rapid and easy fabrication of

multilayer cell-laden tubular hydrogel constructs, and integration with an automated CNC dip-spinning system. The whole manufacturing method has been automatized using microprocessors and software; therefore, potential fabrication protocol can be calculated automatically according to specific numerical models that translate specific mechanical properties into instructions for layers number, fibre orientation(s) and volume fraction(s). This manufacturing equipment has the potential for scale-up and can be adapted to a self-contained GMP manufacturing unit that could even be utilized in the operating theatre or in semi-centralized manufacturing facilities near point of care, attractive from the perspectives of commercialization and clinical translation.

Although patency was only observed after a 12 h period (Fig. 8), the new manufactured SDVG has proven suitable in terms of maximal burst pressure (Supplementary Fig. 6), suture retention (Supplementary Fig. 7), hemocompatibility and blood leak-proof grafting demonstrated by in vivo rabbit models (Fig. 8). The rabbit model was applied here as it has been considered adequate for studies of small diameter vascular conduits due to similarities in thromboplastic and fibrinolytic properties with humans[63]. Although the diameter of the coronary-like SDVG was adjusted in this study to test the grafting capability in a carotid rabbit model, graft wall thickness and mechanical properties were tailored towards the human coronary artery. Suturing conduits of equivalent diameters but with dissimilar wall thickness could result in luminal misalignment between the graft and anastomosed native vessel (Fig. 8d). This is especially true for small conduits of ~1.5 mm in diameter. Misaligned luminal edges generate protruding obstacles for laminal blood flow vectors at the anastomosis, creating recirculation and stagnation zones. According to previous studies, recirculation and stagnation points intensify thrombus formation[64]. Although still conjectural, this could be the reason for the short patency of the SDVG in the rabbit carotid model, considering that even ePTFE vascular prothesis retained patency for longer than 1 week[65]. Another possibility could be associated to exacerbated inflammatory reaction and acute thrombogenesis[66] after implantation of the SDVG, triggered by the presence of trace endotoxin in alginate or methacryloyl gelatin (GelMA)[67]. Notwithstanding that GelMA was carefully prepared and a low endotoxin alginate selected, SDVG constructs were submitted to endotoxin level quantification and an in vivo immunogenicity study utilizing complete SDVG constructs and their individual components (GelMA/alginate/PCL) (Supplementary Fig. 12). Although the endotoxin level present in the SDVG ($3.11 EUmL^{-1}$) is in the range, a previous study reported induction of inflammatory reaction in macrophages[67], and the in vivo immunogenicity results demonstrate a low immune reaction for all individual components of the SDVG and the complete SDVG (Supplementary Fig. 12). Therefore, possible short-term patency due to thrombus formation is in response to an exacerbated immunoreaction cannot be discarded. In this regard, the use of endotoxin-free biomaterials and additional endotoxin control strategies are required for further translation.

Although long-term grafting evaluation is required to demonstrate the efficacy of the bio-inspired SDVG design, these preliminary results show good potential towards clinical usage. Next steps in the development of this SDVG would certainly demand the use of larger and more clinically relevant animal models with experimental follow-up longer than a year[55,68].

In conclusion, this study demonstrated the fabrication of bio-inspired SDVG using the combination of the dip-spinning technique and SBS, which is capable of depositing fibres at defined angles and impart fibre waviness. The mechanical and biological functionality of the scaffold is governed by the presence of the fibres and cells throughout the thickness and length of the graft. Mechanical match to native tissue in each layer was achieved by mimicking fibre angles and tuning vessel thickness, enabling close resemblance to the deformation profile and compliance of human coronary arteries. Specific cell types can be encapsulated to impart biological function to the construct, with homogeneous distributions in the concentric GEAL layers. This study demonstrates potential usability of these vascular grafts based on GEAL layers reinforced with PCL fibres for the replacement of small diameter blood vessels with the prospect of long patency and remodelling toward definitive vascular tissue.

## Methods

**Preparation of GEAL solution.** GEAL was synthesized based on a previously described protocol[69]. Briefly methacryloyl gelatin (GelMA) was synthesized from 10% (w/v) bovine gelatin (Bloom 220, Rousselot, The Netherlands) solution in phosphate-buffered saline (PBS) 1× (10 mM phosphate, 137 mM NaCl, 2.7 mM KCl, pH 7.4), maintained under agitation at 60 °C. Methacrylic anhydride (Sigma, USA) was added dropwise until reaching a final concentration of 8% (v/v) with respect to the volume of gelatin solution in PBS 1×, allowing the functionalization reaction to occur for 3 h. Methacryloyl functionalization was stopped after adding 3 volumes of PBS, 1× followed by dialysis for 7 days (molecular weight cut-off 8 kDa) to remove any un-reacted methacrylic anhydride. The solution was then freeze dried, protected from light and stored at −20 °C for later use. Three stock solutions were prepared. First, the GEAL stock solution was prepared by dissolving freeze-dried GelMA in PBS 1× at 40 °C at a concentration of 20% (w/v). Then, a 2% (w/v) alginate stock solution was prepared by dissolving low endotoxin alginate (PRO-NOVA UP MVG, 4200106, Novamatrix, Norway) in PBS 1× under continuous stirring at 60 °C. A photoinitiator (PI) stock solution was prepared by dissolving 2-hydroxy-4′-(2-hydroxyethoxy)-2-methylpropiophenone (410896, Sigma, USA) in PBS 1× at 85 °C to obtain a concentration of 2% (w/v). Later, the PI solution was maintained at 40 °C temperature before use to prevent crystallization and precipitation. The GEAL solution was obtained after mixing the three stock solutions at defined ratios and the volume was adjusted using PBS 1× to obtain a final concentration of 10% (w/v) of GelMA, 0.5% (w/v) of alginate and 0.2% (w/v) of PI, all with respect to the final volume of the mixed solution. To obtain endotoxin-laden SDVG, low endotoxin alginate was replaced with a medium viscosity sodium alginate (A2033, Sigma, USA), which is known to be high in endotoxin concentrations.

**Deposition of PCL fibre sublayers.** PCL fibre sublayers were deposited using a combination of a custom-made SBS system[30,61] and a dip-spinning machine[29]. PCL, $M_n$ 80,000 (440744, Sigma-Aldrich, USA), was dissolved in a mixture of chloroform/acetone at 80/20 (v/v) to reach a final PCL concentration of 7% (w/v). The SBS configuration is illustrated in Fig. 1a, c. The system comprised of an air compressor (Huracan 1520, Indura, Chile) equipped with a pressure regulator adjusted to 60 psi; additionally, a 10 mL hypodermic syringe driven by a syringe pump (NE-4002X, New Era Pump Systems Inc., NY, USA) to control the injection rate of the PCL solution at $120 \mu L min^{-1}$. Both the compressed air and the PCL solution line are connected into the spraying apparatus, consisting of a concentric nozzle system with a central flow of PCL solution and a concentric peripheral flow of pressurized air (Fig. 1c). The nozzle system can be placed at different positions and angles with respect to the circumferential axis of the deposition rod (or forming vascular graft). For orientated fibre deposition, the system requires a dip-spinning CNC machine that moves a rod up and down at a rate of 138 mm min$^{-1}$ while rotating at 42 rpm (Fig. 1a). This configuration allows a homogeneous fibre deposition along the rotating rod. Rods of specified diameters and lengths were printed using a biocompatible resin (MED610, OBJ04057, Stratasys, USA) and an Objet30 printer (Stratasys, USA). Alternatively, nitinol wire rods of specified diameters can be used. A complete cycle of fibre spinning down-and-up movement takes 30 s and the distance between the SBS nozzle and the point of fibre deposition on the rod surface was kept constant at 30 cm to ensure sufficient solvent evaporation ahead of fibre deposition.

For the fabrication of the adventitia layer, the SBS nozzle was orientated at 67° with respect to the circumferential axis of the graft; for the media layer the orientation was 21°. Fibres were deposited while the rod was subjected to down-and-upward movement. A simultaneous unbalanced sequence of spinning that consisted of 1 s cycles of clockwise rotation at 42 rpm followed by a 0.5 s anti-clockwise rotation at 42 rpm enabled the incorporation of waviness to the oriented PCL fibres (Supplementary Fig. 3). This approach was developed in order to increase the waviness of fibres and imitate the natural configuration of collagen fibres[12,31,32]. To deposit a sequence of PCL fibres in opposite orientations (e.g. +/−21° and +/−67°) and form the angled interleaved PCL sublayer, the spinning movement during fibres deposition was changed to 1 s anti-clockwise rotation at 42 rpm followed by a 0.5 s clockwise rotation at 42 rpm (see Fig. 1). One PCL fibre sublayer consists of one cycle of fibre deposition at a certain angle and

one cycle of fibre deposition at the complementary angle to achieve the interleaved PCL fibre sublayer.

**Deposition of individual GEAL sublayers**. GEAL layers were generated with a custom-made CNC machine[29]. Each layer was fabricated through several dip-spinning cycles of the rod into the GEAL solution previously covered with a PCL sublayer. The GEAL solution was maintained in a water bath at 30 °C to avoid physical gelation at room temperature. Photo-crosslinking was achieved during emergence of the rod from the GEAL solution, by exposing the coated rod to UV light at 365 nm (1.21 W cm$^{-2}$) (using an OmniCure® S2000, Excelitas Technologies, USA). The UV source was placed at a distance of 2 cm from the rod while the coated mandrel was rotating at 42 rpm and emerging at 138 mm min$^{-1}$ upward speed. (Fig. 1b).

The middle and outer graft layer were fabricated by intercalating GEAL and PCL fibre sublayers. PCL fibre sublayers were fabricated using the SBS nozzle oriented at +/−21° and +/−67° with respect to the circumferential axis, respectively (Supplementary Fig. 1). As shown in Fig. 1, one graft sublayer consists of one PCL fibre layer, followed by one photo-crosslinked GEAL layer. Different configurations of sublayers and numbers of graft sublayers (see Fig. 5a) were tested to determine the composition required to fit the mechanical behaviour to human coronary arteries. GEAL sublayers were tested with 1, 2 or 3 cycles of dipping and photo-crosslinking. Additionally, the complete middle and outer graft layer tested comprised a series of 4, 5, 8 or 10 middle or outer graft sublayers, respectively. The best mechanical performing middle layer was fabricated using a series of four middle graft sublayers, with PCL fibres oriented at +/−21° and a GEAL sublayer generated after two cycles of dipping and photo-crosslinking. The best outer graft layer consisted of a series of five sublayers, in which the PCL fibres were oriented at +/−67°, with the GEAL sublayers generated following three cycles of dipping and photo-crosslinking.

In order to visualize the PCL fibre orientations and alignment, spinning rods were prior coated with an alginate hydrogel after dipping the rod into a 2% (w/v) alginate solution (A2033, Sigma, USA) and subsequent ionic crosslinking by dipping into a 5% (w/v) CaCl$_2$ solution (see support information). PCL fibres were then spun onto this crosslinked alginate, which could be easily removed from the rod and facilitated the imaging of PCL fibre orientation in the middle and outer PCL fibre sublayers due to the optic transparency of the alginate (CKX-41, Olympus). The middle and outer PCL fibre sublayers were imaged also using a SEM (LEO 1420 VP). The fibre orientation was measured with ImageJ software (National Institutes of Health, USA). The layer thickness was measured for each sample using a micrometer with 10 μm accuracy.

**Fabrication of a full vascular graft using GEAL and PCL sublayers**. A full bio-inspired SDVG based on the reinforced GEAL hydrogel was fabricated, consisting of three graft layers: inner, middle and outer. For the inner layer, a GEAL sublayer was fabricated after nine dippings and UV-crosslinking cycles using the alginate-coated rod (see Supplementary Methods) and CNC machine. On top of the inner layer, an iteratively improved middle layer was manufactured as described above. Subsequently, the concentric-outer graft layer was fabricated atop of the middle graft layer. Finally, the rod and alginate coating were mechanically separated. The alginate coating prevented scathing of the inner or luminal layer on removal from the rod. The whole manufacturing process was performed under sterile conditions within a biosafety cabinet, including preparation of the GEAL solution. After chemical modification, and initially during dialysis, the first hour of dialysis was performed using a water solution supplemented with 1% (v/v) chloroform for sterilization. The prepared GEAL mixture was then submitted to three cycles of heating and cooling (20 min at 70 °C and 20 min at 4 °C). Additionally, bioburden testing (Inoculation of SDVG extract in Blood agar, Sabouraud Dextrose Agar, Valtek, Chile) and mycoplasma testing (MycoAlertTM Mycoplasma Detection Kit, Lonza) were routinely performed for fabricated SDVG. SDVG pieces ~2 mm² were incubated in PBS 1×, pH 7.4 (1 g of SDVG in 5 mL of PBS) for 24 h at room temperature, followed by vigorous agitation and centrifugation (500 × $g$ for 5 min), and the supernatant was tested for bioburden and mycoplasma. SDVG constructs used for experiments with cells or animal models were treated with 1% (v/v) penicillin–streptomycin (15140-122, Gibco, USA) in culture media for 24 h at 37 °C.

**Middle and outer layer tensile testing**. The middle and outer layer of the bio-inspired SDVGs containing PCL fibre/GEAL sublayers were tested in uniaxial tension using a Texture analyser (Stable Micro Systems, TA.XT.plus, Surrey, UK) equipped with a 5 N load cell. Rectangular samples of the construct were tested to determine their circumferential and longitudinal stress–strain profiles. Three samples were cut and tested in both directions, circumferentially and longitudinally for each layer. Sample thickness and width were measured using a micrometer with 10 μm accuracy. For preconditioned samples, the constructs were subjected to five loading and unloading cycles of applied strain at a constant rate of 10 mm min$^{-1}$. The preconditioning loading/unloading cycles of the outer graft layer in the longitudinal and circumferential direction were conducted to a strain level of 13% and 30%, respectively. Differences in strain level were iteratively adjusted to the desired target J-shaped strain–stress curves matching human coronary arteries. In

the case of the middle graft layer, the strain for preconditioning was 35% and 30% in the circumferential and longitudinal tensile testing, respectively. Uniaxial testing for both circumferential and longitudinal samples was performed at a constant rate of 10 mm min$^{-1}$ (Supplementary Fig. 2a).

**SDVG tensile testing**. Bio-inspired SDVGs consisting of PCL fibre/GEAL sublayers were tested in uniaxial tension using a universal testing machine (Instron 3342, Norwood, MA, USA) equipped with a 10 N load cell. Rectangular samples oriented circumferentially and longitudinally were cut, maintained and tested under in vitro conditions by immersion in PBS 1× at 37 ± 0.5 °C. For each bio-inspired SDVG, five samples were tested for each tensile direction. Sample thickness and width were measured using a micrometer. Preconditioning step was conducted by tensile loading (longitudinal and circumferential) with five loading/unloading cycles performed using a maximum strain of 30% applied at a constant rate of 10 mm min$^{-1}$. The strain value was adjusted to obtain a better fit of the SDVG stress–strain curve with human coronary arteries. Uniaxial tensile testing was performed at a constant strain rate of 1 mm min$^{-1}$ and up to 30% strain (Supplementary Fig. 2b).

Stress–strain curves for all experiments were obtained from axial loading and clamp displacement data recorded during the test. Stress was computed as $F/A$, in which the $F$ corresponds to the tensile load with a precision of 0.01 N and $A$ is the initial cross-sectional area. Strain was calculated as $100 \times L/L_0$, with $L$ and $L_0$ as the current and initial sample length, respectively.

**Pressurization tests**. Pressurization tests were conducted to evaluate the mechanical behaviour of the complete bio-inspired SDVG under simulated in vivo (human) loading and pressure conditions[70]. Tests were performed in a customized set up using a universal testing machine (Instron 3342, Norwood, MA, USA), adapted with a plastic transparent chamber filled with PBS 1 × and maintained with a controlled temperature of 37 ± 0.5 °C. Internal pressure was applied via an auxiliary line of PBS at 37 °C connected to the internal graft lumen. The tubular samples were attached to insertion tubes with instant glue and Teflon tape. The test was filmed to track changes in diameter. Pressure was measured at the entrance of the chamber via a pressure transducer, and the graft diameter was measured from the video using image processing techniques; the diameter was measured along the length of the sample and averaged for each pressure. Five tubular sample average lengths of 3 cm were tested. Before testing, five loading/unloading cycles in the axial direction of the graft were performed, applying a maximal strain of 30% at a constant rate of 10 mm min$^{-1}$ as a preconditioning step. An additional pre-conditioning step in the circumferential direction was performed using five cycles of pressurization from 0 to 200 mmHg. Pressurization tests were conducted on SDVGs with three different constant axial strains, 10%, 20% and 25%, all separately.

The compliance value of the full bio-inspired SDVGs (%$C$) was computed from the experimental data over three pressure ranges (50–90, 80–120 and 110–150 mmHg), according to standard ISO 7198 (ANSI/AAMI/ 2010), and using the following equation:

$$\%C = \frac{(R_{P_2} - R_{P_1})/R_{P_1}}{P_2 - P_1} \times 10^4,$$ (1)

where $P_1$ and $P_2$ correspond to the lower and higher pressure values of the range in mmHg, and $R_{P_1}$ and $R_{P_2}$ are the external radiuses generated at the respective pressures.

**Cell culture**. HUVECs (ATCC® CRL1730™) and BM-MSCs (expanded until passage 5) (#PT-2501, Lot# 0000423370, Lonza, USA) were cultured and expanded in high glucose Dulbecco's modified Eagle's medium (16000-044, Gibco, USA) supplemented with 10% (v/v) fetal bovine serum (16000-044, Gibco, USA), 2 mM glutamine (25030-081, Gibco, USA) and 1% (v/v) penicillin–streptomycin (15140-122, Gibco, USA), and incubated at 37 °C, 5% CO$_2$ and 96% of humidity. Cell cultures before experiments and cell storage were routinely subjected to mycoplasma testing (MycoAlertTM Mycoplasma Detection Kit, Lonza) using culture supernatants as testing samples. HUVECs or BM-MSCs were mixed in the GEAL solution at a concentration of 10 million cells mL$^{-1}$. SDVGs were fabricated as described above using a mixture of GEAL solution and cells in order to encapsulate the cells within the GEAL sublayers. Media changes were performed every 2 days. Cell culture of complete or sectioned SDVGs were performed similarly, except that the culture media were additionally supplemented with 1× amphotericin B (15290-026, Gibco, USA). This preventive measure was performed to mitigate exposure to microorganisms post fabrication.

**Live/Dead assay and Ki-67 staining**. After fabrication, 5 mm cylindrical section lengths of the bio-inspired and cell-laden SDVGs was maintained under cell culture conditions (see previous section). SDVG cell cultures were terminated at different time intervals (days 1, 7, 14, 21 and 28), and in some cases, individual graft sublayers were carefully peeled off with the aid of a stereomicroscope for visualization. SDVG samples and sublayers were washed in 1× PBS and stained to evaluate cell viability using the LIVE/DEAD® Viability/Cytotoxicity Kit (L3224, Thermo Fisher, USA) according to the manufacturer's guidelines. Briefly, SDVG

sections and single sublayers were incubated with 500 µL of LIVE/DEAD® reagent (4 µM ethidium homodimer-1 and 12 µM calcein) for 45 min at room temperature with gentle agitation and protected from direct light. Later, sublayers were mounted in PBS for visualization in a confocal microscope (Sp8, Leica). Image analysis was conducted using the LAS X software (Leica Microsystems) and ImageJ software (NIH). Additionally, sublayers obtained from cultured SDVGs at 28 days were fixated using 4% paraformaldehyde for 20 min at room temperature. After several washes in 1× PBS, samples were permeabilized and blocked using 1% (w/v) bovine serum albumin, 0.2% (w/v) TX-100 in 1× PBS for 30 min at room temperature. For Ki-67 detection, samples were incubated with anti-Ki-67 (1/1000) overnight at 4 °C (ab15580, Abcam) in a humidified chamber. These samples were then incubated with the secondary antibody, goat anti-rabbit DyLight® 594 (#35561, Thermo Fisher) and DAPI (4′, 6-diamidino-2-phenylindole, Thermo Fisher) for nuclear staining. Coverslips were mounted in FluorSaveTM (Merck Millipore) prior to analysis.

**Histological analysis**. On the same day of fabrication, bio-inspired SDVGs were embedded in optimal cutting temperature compound (Tissue-Tek) and transversally cryo-sectioned to 14 µm using a cryostat (Microm, HM525, Walldorf, Germany). Cells were stained by incubating the samples in Hoechst 33342 solution (Thermo Scientific, USA) following the provider's protocol. Transversal cuts were visualized using a fluorescent microscope (CKX-41, Olympus, USA). Additionally, surgically explanted SDVGs from rabbit carotid graft models were prepared for H&E histological studies. Briefly, SDVGs were fixed in 10% formalin solution (HT501128, Sigma, USA) overnight at 4 °C, dehydrated using a series of increasing alcohol concentrations and xylene, and embedded in paraffin using a Tissue-Tek®TEC Embedding system (Sakura, Tokyo, Japan). Tissue sections of 5 µm in width were obtained using an AccuCut®SRM microtome (Sakura, Tokyo, Japan). Later, slides were rehydrated with alcohol and increasing concentrations of water, and stained in Harris haematoxylin (MEDITE GmbH, Burgdorf, German) for 30 s and alcoholic Eeosin Y (HT110116, Sigma, USA) for 5 s. Finally, the samples were dehydrated and mounted with Entellan (#107960, Merck, Germany) for visualization.

**Platelet activation and clotting time evaluation**. To unveil the potential of thrombogenic induction of the SDVGs due to blood–graft interface contact activation, two separate assays were performed, a platelet activation assay[45] and clotting assay[71]. For the platelet activation assay, 10 mL blood samples from three healthy human volunteers were extracted in sodium citrate blood collection tubes (BD Vacutainer, 369714, Becton, Dickinson and Company, USA) after signing a written informed consent, previously approved by Institutional Ethical Committee. Subsequently, PRP was obtained by centrifugation at $80 \times g$ for 20 min at room temperature, after which PRP samples were treated carefully to avoid shaking-induced activation. PRP samples each 200 µL were transferred into 96-well plates (Falcon® 96-Well Cell Culture Plates, 351172, Corning®, USA), and incubated in the presence of equivalent surface area of SDVG segments or fabrication elements of the SDVGs. Experimental groups were: PRP incubated over a BM-MSCs monolayer, PRP + crosslinked 10% (w/v) GelMA solution, PRP + crosslinked 10% (w/v) GelMA solution with encapsulated BM-MSCs at $10^7$ cells mL$^{-1}$ concentration, PRP + acellularized SDVG and PRP + BM-MSCs-laden SDVG. Incubations were conducted for 1 h at 37 °C under orbital agitation (80 rpm). As positive platelet activation control, PRP was incubated for 10 min in the presence of 100 ng mL$^{-1}$ of 4β-phorbol 12-myristate 13-acetate (P8139, Sigma-Aldrich, USA), and non-activation control, PRP was incubated in the presence of equivalent samples volume of citrate buffer 0.1 M for 1 h at 37 °C under agitation. After incubation, 50 µl of PRP was collected for flow cytometry analysis. Platelets were stained for 20 min at room temperature in the dark using anti-human antibodies CD42a (Clone ALMA. 16), CD63 (Clone H5C6) and CD62P (Clone AK-4) all from BD Bioscience, conjugated to the fluorochromes FITC, PE and APC, respectively. Data acquisition was performed on a FACSCanto II™ using the FACS Diva software (Becton Dickinson, CA, USA) and data were analysed using the FlowJo software (Tree Star, Canton, OH, USA). For evaluation of clot formation induced by the luminal surface of fabricated SDVG, the method described by Motlagh et al.[71] was conducted. Blood was extracted in sodium citrate blood collection tubes from healthy volunteers. First tube was eliminated to avoid possible contamination with tissue thromboplastin. Next, in order to activate clotting, 500 µL of CaCl₂ (0.1 M) was mixed with 5 mL of blood immediately prior to the assay. In parallel, 2 cm tubular fragments of each tested graft were added to a 12-well plate in quadruplicate, along with controls: (1) empty wells, (2) acellularized SDVGs, (3) SDVGs + BM-MSC ($10^7$ cell mL$^{-1}$), (4) commercial vessel (GORE-TEX® vascular graft #RRT0807007OL), (5) broken microscopy coverglass (9mm² pieces of glass to accelerate coagulation) and (6) CaCl₂-free blood (as coagulation negative control). Activated blood (100 µL) was added over the inner luminal surfaces of the SDVG and commercial synthetic graft, including the surfaces of fragmented coverglass and empty polystyrene wells, and incubated for 5, 10, 15 and 30 min at room temperature. After each time point, 3 mL of distilled water was added and incubated for 5 min at room temperature in order to lyze free red blood cells that did not form part of the clot. Two hundred microliters of each sample was then transferred to a 96-well plate and absorbance (540 nm) was measured using a plate reader (NanoQuant Infinite TECAN M200 PRO). Absorbance is proportional to

free haemoglobin, and consequently, inversely proportional to the clotting induction of each surface sample.

**In vivo grafting study in a carotid rabbit model**. A preliminary SDVG grafting study was conducted using six female New Zealand white rabbits weighing ~4 kg each, at the Universidad de los Andes Animal Facility (Santiago, Chile) following the institutional guidelines for care and experimentation with laboratory animals, with Institutional Ethical Committee approval (Ethical Committee, University of the Andes, presided by Juan Eduardo Carreño, Ph.D. M.D.). Two rabbits were used as surgical controls, in which carotid incision and anastomosis was conducted to cut and restore carotid circulation. Additionally, two groups of two rabbits each were utilized to evaluate circulation grafting for acellular and cellularized (BM-MSCs) SDVGs. Rabbits were anaesthetized using an intramuscular injection of 30 mg kg$^{-1}$ of ketamine and 3 mg kg$^{-1}$ of xylazine. The rabbits were then maintained anaesthetized by inhalation of vaporized sevoflurane (2.5%). Prior to surgery, heparin was added at 150Ukg$^{-1}$ of animal as an anticoagulation agent. After ventral neck shaving and sterilization, middle incision and carotid isolation were performed. Subsequently, the carotid was proximally and distally clamped, cut and a SDVG anastomosed in between the clamps. Anastomosis of the 2 cm length and 1.5 mm diameter SDVG was performed in an end-to-end fashion using approximately eight sutures (Prolene 8/0, Ethicon). After blood flow restoration, the wound was flushed with gentamicin (200 U mL$^{-1}$) and closed using a 5/0 monofilament nylon suture (Ethicon). Post surgery, rabbits were treated for 4 days with daily doses of 0.5 mg kg$^{-1}$ of meloxicam, 10 mgkg$^{-1}$ of enrofloxacin and 2 mg kg$^{-1}$ of aspirin. Graft patency was monitored at 12 and 36 h post surgery, and every 10 days using a colour Doppler ultrasound device (SonoSite, M-Turbo, Fujifilm, USA). After 30 days, rabbits were euthanized by an overdose of ketamine/xylazine and the SDVGs explanted for further studies.

**Quantification of endotoxin in the fabricated SDVG**. Two SDVGs (6 cm in length) were cut into 4mm² pieces and incubated in separate 15 mL conical tubes with 1 mL of 1× PBS, pH 7.4, at 37 °C for 24 h. An Endosafe® endotoxin testing system (Endosafe®-PTS, Charles River, USA) was conducted using the 1× PBS supernatant after centrifugation ($500 \times g$ for 5 min) and dilution (1:20) in sterile and endotoxin-free dH₂O. The concentration of endotoxin in the eluent (1× PBS) was quantified using the Endosafe® LAL Cartridges (#PTS2005F, Charles River, USA) and the portable system Endosafe®-PTS (Charles River, USA) and expressed in EU mL$^{-1}$.

**Statistical analysis**. Data are presented as mean ± SD or mean ± SEM, also indicated in the figure caption. Statistical significance was determined using two-tailed Mann–Whitney $U$ test in compliance of bio-inspired SDVGs (Supplementary Table 1). Two-tailed Mann–Whitney $U$ test was also applied for in vivo immunosuppression functional experiments in the mouse model, platelet activation assay and clotting time assay (Supplementary Fig. 10, Fig. 8a–c, respectively). All statistically analysed data comply with the non-parametric Mann–Whitney $U$ test. Platelet activation assay was performed utilizing three human blood samples from different donors and conducted in experimental triplicates, whereas clotting time assay was done using four human blood donors in experimental triplicates. Sample size calculations were determined using the online resource from IACUC of Boston University (https://www.bu.edu/researchsupport/compliance/animal-care/working-with-animals/research/sample-sizecalculations-iacuc/)[72], with a power level of 95% and estimated variance from previous publications. Specifically for compliance experiments, estimated variance was determined based on Claes et al.[34] and van Andel et al.[35], whereas the immunocompetent mice model was based on Campos-Mora et al[73]. All comparison groups are considered to have similar data distributions. In the implantability study in arterial circulation using a rabbit model, results are presented as a preliminary study, and therefore only two animals per group were performed and no statistical analysis is presented.

All experimental outcomes were obtained after blinded quantitative analysis of samples results. For all statistically analysed experiments, 95% of confidence was used and significance was denoted as *$p \leq 0.05$, **$p \leq 0.01$ and ***$p \leq 0.001$. $n$ values for each experiment are given in the figure caption.

**Reporting summary**. Further information on research design is available in the Nature Research Reporting Summary linked to this article.

## Data availability
The data that support the findings of this study are available from the corresponding author upon reasonable request. Additionally, the data that are displayed in the graphs of Fig. 5c–e are available in figshare with the identifier https://doi.org/10.6084/m9.figshare.8298449, and the data displayed in the graphs of Fig. 6a, b and Fig. 6d–f are available in figshare with the identifier https://doi.org/10.6084/m9.figshare.8298482.

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

## Acknowledgements

We want to thank all the researchers in the Cells for Cells, Dr. Valenzuela (Pontificia Universidad Católica de Chile), the Bio-Active Materials group (The University of Manchester, UK) and Dr. García's (Universidad de Santiago de Chile) laboratories for insightful discussions and suggestions. We thank Dr. Gowsihan Poologasundarampillai for assistance with synchrotron-CT and the Diamond Light Source for beamtime (MT15507) on the I13 Diamond-Manchester Branchline and the staff there for all of the assistance with data collection. This work was made possible by the facilities and support provided by the Diamond-Manchester Collaboration and the Research Complex at Harwell, funded in part by the Engineering and Physical Sciences Research Council (EPSRC) (EP/M023877/1). We acknowledge financial support from CORFO 18COTE-97983, FONDEF IDEA I15I10545, CONICYT by the grants FONDECYT Nos 3160680, 1170608, FONDEQUIP EQM130028 and Newton-Picarte REDES No. 140144 and the Universidad de los Andes PMI program UAN1301.

## Author contributions

T.L.A. and J.P.A conceived the idea. J.J.B. contributed idea and development of SBS for the deposition of fibres at defined angles to control mechanical properties, J.J.B. and C.A.W. contributed in part to the idea of intercalating cell-laden hydrogels with reinforcing PCL fibres. T.L.A. designed and experimentally realized the study. J.P.A supervised the entire project and is responsible for the infrastructure and project direction. C.T. and C.A.W. designed the in vivo mouse model experiments and C.T. experimentally realized the study and analysed the data. C.A.W. contributed with the immunological analysis. The SBS was contributed by J.J.B. who co-wrote the paper and supervised in part C.A.W. and T.A. during their visits to The University of Manchester. J.M. conducted X-ray CT and synchrotron experiments, optimized scanning parameters, analysed tomography reconstructions, performed data and image analysis to determine construct porosities. C.M.G.-H. supervised the mechanical studies and analysis. J.E. contributed with super-vision and laboratories for biomaterials preparation and quality testing. C.S. performed SDVG manufacturing for burst pressure mechanical testing and in vivo experiment in the rabbit model. F.V. performed anastomosis surgeries in the rabbit model, patency analysis and gave insightful information concerning the surgical performance of the SDVG. M.O. assisted the surgical procedure, executed surgical post treatment and animal maintenance according to international guidelines for care and experimentation with rabbit models. M.O., C.S., C.T. and G.Z. assisted the surgical procedure in the rabbit model. G.Z. conducted the H&E experiments, clotting time assay and live/dead experiment using confocal microscopy. L.M.V. supervised the whole procedures and co-wrote the paper. M.K. contributed in the immunological analysis and supervision of the project. T.L.A., J.P.A. and L.M.V. prepared the manuscript. All the authors discussed the results, commented on and revised the manuscript.

## Additional information

**Competing interests:** M.K. is the CSO of Cells for Cells and Regenero, J.P.A. and M.O. received stipends from Cells for Cells. The other authors declare no competing interests.

