## [Peer Review File · Nature Communications]

Reviewers' comments:

Reviewer #1 (Remarks to the Author):

The authors are commended for devising a series of processing methods to approach the multilayered design of the artery with control of the stress-strain curve. In this respect, the accomplishment seems substantial. However, for impact in the field, sufficient strength is required, and these constructs lack it: there is no explicit mention of "strength" that I could find in terms of results presented or how the approach could be modified to achieve it (not that the latter in and of itself would warrant publication in NCOMMS). But inspection of the stress-strain curves, assuming they are plotted to failure, shows the UTS is ~200KPa, about a factor of 10 below the physiological value. Even if 2MPa were achieved, physiological burst pressure would have to be demonstrated to show potential. And for publication in NCOMMS, I would expect beyond that a successful implantation study in the arterial circulation, not subcutaneous as presented here.

Reviewer #2 (Remarks to the Author):

This is an intriguing paper that reports a process for the construction of replacement arteries that effectively mimic the mechanical characteristics of native arteries. The authors describe their innovative fabrication process and show that cells included in the resulting vessels continue to be functional. While the reported methods and results are exciting, there are numerous problems with grammar, word choice, and punctuation that sometimes make it difficult to understand the authors' intent; aside from this, the paper is outstanding. Specific issues to be addressed are described below.

Many of the figures (see 3, 4, and 5) and their associated captions are confusing since the authors don't specify the meaning of all of the curves included in the plots and/or the caption descriptions don't seem to accurately correspond with the content of the plots. These need to be checked and corrected for better comprehension. Also

On line 174, the authors describe a pre-stretching step required to produce the characteristic J-shape of arteries, though little detail is given. It is not clear whether this was done during the fabrication itself, on a layer-by-layer basis as the vessel was formed, or just at the end of the fabrication process. This needs to be clarified. The paper should also specify and justify the extent of the pre-stretch utilized.

It is not clear why the authors chose to perform uniaxial strip tests on the fabricated vessel material, instead of simply testing biaxial behavior. This should be justified.

Table 1 does not appear to be referenced in the text of the manuscript

It is not clear that the cells used in the fabrication are relevant. Why weren't vascular cells (i.e. endothelial cells, smooth muscle cells, fibroblasts) used?

Please describe the details of the vessel culture from days 1 to 7.

Cell survival appears to be equated to function, but it should be made clear that vessel function (e.g. contractility) was not tested in this paper.

SIVG does not appear to be defined anywhere in the paper

Reviewer #3 (Remarks to the Author):

Title: Rapid fabrication of reinforced and cell-laden vascular grafts structurally inspired by human coronary arteries.

Authors: Akentjew, T.L., et al.

General Comments:

This is an interesting paper describing a combination of spinning and hydrogel technologies, that can deposit concentric layers of cell-laden hydrogels that are interspersed with re-inforcing fibers to provide mechanical strength.

Cell layers are contained in a methacryloyl gelatin-alginate (GEAL) substrate, while the reinforcing fibers are made from slow-degrading polycaprolactone (PCL). The idea of interspersing mechanically reinforcing fibers within a living tissue is novel and potentially important. However, this paper has multiple drawbacks that limit enthusiasm substantially. Overall, important information is lacking on the process, and the figures contain many errors and substantially lack clarity. Missing important information includes: what is the diameter of the grafts? How long does the graft manufacturing process take – can cells survive this period? Is the process sterile? How big are the PCL fibers? This is difficult to infer from the paper. What is the phenotype or appearance of the cells in the construct? There is zero histological information. How were MSC harvested and characterized? No information on this either.

Also, there is overall a lack of care in preparation of the paper. The figure legends contain many errors, the statistical analysis is uneven and poor overall, and the Methods descriptions do not comport with the data in the paper. Overall, English grammar should be improved as well.

Specific Comments:

1. Line 123: the statement is made that the GEAL solution is pre-crosslinked, but then the authors state that the dipped layer is exposed to UV light to crosslink the material. please clarify.
2. The phrase "circumferential axis" is used throughout the paper and is unclear to this reviewer.
3. After spraying the PCL layers, do the fibers fuse at the points of contact? What is the data for or against fiber fusion, which will significantly impact the vessel mechanical properties? What is the fiber diameter, is it uniform, how is it controlled?
4. The legend for Figure 4 is very confusing – panels a-c are stated as having data in the grey lines, but these lines are identical across the images, and the dotted lines are not described at all. It is pretty impossible to tell what this figure is saying. Do the authors have data to show on actual stress-strain curves of native human coronary arteries?
5. Similarly the legend for Figure 5 makes no sense – the grey lines and solid squares that are talked about in the legend simply aren't there in the figure. And, panels b-e contain multiple black lines of unclear significance- if these are replicates, then should not some error bars be included or something? In Figures 5 and 7, where is the "grey region of the range of the model of the mechanical properties" derived from? This is not at all clear to the reader, and serves only to confuse.
6. In Table 1, what is the "n" value for these data? This bears on the statistical significance and differences.
7. Figure 8 needs scale bars. What is the meaning of the blue color – does this signal live or dead cells or proliferating cells? What are the dimensions of the construct? The reagent used does not show cell replication per se – it appears to show mitochondrial activity??
8. Figure 9 shows cells counts from histology, but there are no histological images shown in the main paper. This is very disappointing.
9. In the Methods section, the Histology paragraph talks about skin transplants and positive controls for immunity, and talks about FACS analysis using different markers for immune cells, but none of this data is shown in the paper. It appears as if this text was taken from another manuscript, since it does not match up with the data in this paper?

Reviewers' comments:

Reviewer #1 (Remarks to the Author):

The authors are commended for devising a series of processing methods to approach the
multilayered design of the artery with control of the stress-strain curve. In this respect, the
accomplishment seems substantial. However, for impact in the field, sufficient strength is
required, and these constructs lack it: there is no explicit mention of "strength" that I could
find in terms of results presented or how the approach could be modified to achieve it (not
that the latter in and of itself would warrant publication in NCOMMS). But inspection of the
stress-strain curves, assuming they are plotted to failure, shows the UTS is ~200KPa, about a
factor of 10 below the physiological value. Even if 2MPa were achieved, physiological burst
pressure would have to be demonstrated to show potential. And for publication in NCOMMS,
I would expect beyond that a successful implantation study in the arterial circulation, not
subcutaneous as presented here.

Response: We agree with the reviewer and have conducted experiments to assess questions
about the strength characterization and evaluation of physiological burst pressure of the
constructs. In these regards, sufficient ultimate tensile strength (UTS) was evaluated by
longitudinal tensile testing until fracture of the small diameter vascular graft (SDVG), whereas

for physiological burst pressure assessment pressurization testing of SDVGs were conducted
until failure.

In our first draft, we initially performed tensile tests using a fixed upper limit of strain to
evaluate mainly the stress-strain J-shape curve, without recording the UTS. Therefore, as well
mentioned by the reviewer, a reader could assume that results were plotted to failure. This is
now clarified in the text and material and methods. In the main text, Line 313 the following is
stated: “The construct exhibited mechanical response similar to human coronary arteries,
both in longitudinal and circumferential directions under uniaxial tensile testing up to a 30%
strain range (Fig. 7a-b).”, and additionally, in material an methods line 781, the following is
mentioned: “Uniaxial tensile testing was performed at a constant strain rate of 1 mm/min and
up to 30% strain (see Fig. S2b).” However, since the failure data is relevant for the application
of this technology, as stated by the reviewer, we included information about the tensile
testing

to failure and physiological burst pressure results in the main text and supporting material.
We have summarized these results in line 315 for the resistance until failure: “Longitudinally
tested SDVG exhibited a maximum failure strength of 520 ± 56 kPa, which is in the range of
coronary arteries of individual above 35 years old^{18,45} (Fig S4)”. In line 363 the following results
concerning burst pressure were included: “Due to safety and clinical concerns, vascular grafts
must also meet adequate burst pressures and suture retention strength^{25,47}.The fabricated
full SDVGs exhibited burst pressures of 1630 ± 180 mmHg, similar to values reported for
human saphenous vein, as well used as autologous graft for coronary bypass⁴⁸ (see Fig S5).”.
Being the suture retention strength a particularly important aspect of vascular grafts in
regards to safety and clinical practice, we have included in line 366 additional results of suture

retention strength: “Suture retention strength was determined at 143 ± 13 grams-force for
SDVGs, similar to reported values for human internal mammary artery (IMA)⁴⁷(see Fig S6)”.

We agree with the reviewer, concerning the need of successful implantation study in the
arterial circulation. To address this issue, we have included a whole new chapter in line 501
title “Implantability study in arterial circulation using a rabbit model”. A preliminary study for
artery grafting was conducted in a rabbit carotid model (1-month period and patency test)
using a total of 6 rabbits, two with artery to artery anastomosis as surgery control, two with
acellularized SDVG and two with BM-MSCs-laden SDVG. During surgery, the grafting
experiments showed good suturing performance, no blood leakage observed, absence of
apparent immunological rejection or inflammation of the implanted grafts and absence of any
unexpected and adverse events during grafting. Except for the anastomosis control all
implanted rabbits showed SDVG patency between 12 h and 36 h. Grafting results are
discussed in line 623-640. We clarified in the discussion too that our SDVG design and iterative
improvement was originally directed to human coronary arteries. Nevertheless, the rabbit
model is an appropriate model to evaluate adverse effect derived from thrombosis and
immunorejection. In order to evaluate the SDVG in a rabbit carotid model, we modified the
SDVG design in terms of diameter, but not wall thickness and mechanical properties to match
that of the rabbit carotids. Therefore, the rabbit carotid model has certain limitation to assess
the grafting and bypass potential of the new SDVG, which are discuss in the manuscript (Lines
625-640). Additionally, in the revised version of the manuscript, we discuss the necessity for
testing on larger animal models, though we deem such tests out of the scope of the current
work, and indeed such tests will require significant further funding and time.

Additional to the implantability study, we have added *in vitro* hemocompatibility results (lines
451-499) and suturing retention strength (previously addressed in this response) to further
support arterial grafting potential.

Taking into consideration the issue raised with respect of the subcutaneous implantation by
the reviewer, we have further clarified the objective of that experimentation in the actual
manuscript. For example, in lines 586-593 ("Cells remained viable during fabrication, and
demonstrated proliferative and functional capacity within the SDVG (Fig. 8 and 9). In vivo
immunogenicity experiments demonstrate that the encapsulation of BM-MSCs lowers the
rejection and inflammation response cause by the subcutaneous implantation of an
endotoxin-laden SDVG. The immunosuppression cell function is of importance in the design of a
new vascular graft, considering a cause of graft failure is associated to chronic inflammatory
response⁵. These results prove the importance of fabricating MSC-cellularized grafts to
prevent rejection and enhance the functionality of the implant by promoting cell repopulation
and definitive tissue remodeling after implantation⁷⁴.") we discuss that the subcutaneous
implantation model was used to evaluated the functional competence of encapsulated BM-
MSCs, suggesting that cells are still functionally viable after the manufacturing process, but as
well to demonstrated that BM-MSCs in the SDVG design could ameliorate inflammatory
responses derived from graft implantation, which has been associated as one cause of
vascular graft failure in clinic.

Reviewer #2 (Remarks to the Author):

This is an intriguing paper that reports a process for the construction of replacement arteries

that effectively mimic the mechanical characteristics of native arteries. The authors describe
their innovative fabrication process and show that cells included in the resulting vessels
continue to be functional. While the reported methods and results are exciting, there are
numerous problems with grammar, word choice, and punctuation that sometimes make it
difficult to understand the authors' intent; aside from this, the paper is outstanding. Specific
issues to be addressed are described below.

Many of the figures (see 3, 4, and 5) and their associated captions are confusing since the
authors don't specify the meaning of all of the curves included in the plots and/or the caption
descriptions don't seem to accurately correspond with the content of the plots. These need to
be checked and corrected for better comprehension.

Response: We thank the reviewer for his helpful comments. All figures and figure captions
have been revised accordingly. We have revised the entire manuscript for improved clarity
and to address the English throughout.

Also on line 174, the authors describe a pre-stretching step required to produce the
characteristic J-shape of arteries, though little detail is given. It is not clear whether this was
done during the fabrication itself, on a layer-by-layer basis as the vessel was formed, or just at
the end of the fabrication process. This needs to be clarified. The paper should also specify
and justify the extent of the pre-stretch utilized.

Response: We thank the reviewer for pointing this out. We have clarified the rationale behind
and use of the pre-stretching step, hereon called preconditioning step. We now include in line
206 the following: "In an effort to match the J-shape stress-strain curves¹⁸, a series of 24 PCL

fibre sublayers were deposited at +/-21° and preconditioned after fabrication by stretching
cycles before subjecting the construct to longitudinal tensile testing.”, and in the materials
and methods section in line 767 the following was included: “The preconditioning
loading/unloading cycles of the outer graft layer in the longitudinal and circumferential
direction were conducted to a strain level of 13% and 30%, respectively. Differences in strain
level were iteratively adjusted to obtain the target J-shape strain-stress curves. In the case of
the middle graft layer, the strain for preconditioning was 35% and 30% in the circumferential
and longitudinal tensile testing, respectively. Uniaxial testing for both, circumferential and
longitudinal samples, were performed at a constant rate of 10 mm/min (see Fig. S2a).”.

It is important to understand the effect of angled fibre deposition on mechanical properties in
each of the layers and their pre-conditioning’ tests were devised to assess this at the level of
the outer graft layer, middle graft layer and complete SDVG construct, as clarified in the text
(Lines 234-236: “In order to define the structural configuration of the middle and outer graft
layers of the new bio-inspired small diameter vascular graft (SDVG), iterative testing of
differently layered constructs was performed in an effort to match the stress-strain profiles of
the media and adventitia layers of human coronary arteries¹⁵” and lines 311-315: “Nonlinear
and anisotropic mechanical response are maintained when tested together in the
configuration of a full SDVG, comprising of inner, middle and outer graft layers, GEAL
sublayers, with wavy, orientated PCL fibre sublayers and preconditioning. The construct
exhibited mechanical response similar to human coronary arteries, both in longitudinal and
circumferential directions under uniaxial tensile testing up to a 30% strain range (Fig. 7a-b).”)
Briefly, for the outer, middle graft layer and complete SDVG formulations, prior to mechanical
testing and after fabrication, layers were submitted to the pre-conditioning step (5 cycles of

30% strain). Beyond this, we have conducted pressurization testing, in which SDVG were
previously submitted to 5 cycles of 30% strain in the longitudinal direction after being
mounted cylindrically in the pressurization system coupled to the tensile machine (see
material and methods line 796) and 5 cycles of 200 mmHG pressure loading prior to test. In an
*in vivo* or clinical implantation setting, the pre-conditioning step may be obviated, although
this has not been addressed in the actual study (line 548: “It may even be possible to conduct
the preconditioning in situ on implantation by pulsatile blood pressure without need of
previous preconditioning and is the subject of future studies.”).

It is not clear why the authors chose to perform uniaxial strip tests on the fabricated vessel
material, instead of simply testing biaxial behavior. This should be justified.

Response: Indeed, biaxial mechanical testing would be more appropriate for mechanical
analysis of vasculature tissues, as mentioned by the reviewer, especially as this type of tissue
is submitted to biaxial stretching during pulsatile blood flow. However, and since our study
strongly rely on previous experimental and theoretical studies based on uniaxial testing
analysis, we decided to use equivalent experimental setting to guide our fabrication design in
order to match the native mechanics.

Table 1 does not appear to be referenced in the text of the manuscript

Response: We thanks the reviewer for pointing this out. Table 1 is now referenced in the new
manuscript in line 354 Quoted here: “Compliance values for SDVGs at 10% and 20% of axial
stretching condition showed no statistical difference with reported results for human

coronary arteries (Table 1)."

It is not clear that the cells used in the fabrication are relevant. Why weren't vascular cells (i.e.
endothelial cells, smooth muscle cells, fibroblasts) used?

Response: In fact, we did use endothelial cells, HUVECs cells (human umbilical vein
endothelial cells), which are known to be more oxidative-stress and hypoxia sensitive cells
than mesenchymal progenitor cells, therefore, if HUVECs are responding appropriately to the
manufacturing technique in terms of viability, where hypoxic and free radicals are the main
source of possible cell damage, we can argue with more certainty that our methodology is cell
friendly or cell compatible. In regard to bone marrow mesenchymal stem cells (BM-MSCs),
they have been chosen as the cell source in the cellularized SDVG design for being a less
invasive source, easily expanded, immune-evasive allogenic cell type, capable to differentiate
to vascular tissues and known to confer positive grafting results in previous vascular graft
developments. We have now clarified the rationale behind all cell types used in the
fabrication, *in vitro* and *in vivo* studies.

The rationale for the selection of HUVECs for the proliferation and LIVE/DEAD assay is now
clarified in line 381 ("HUVECs were chosen for this study as they have been identified as more
oxidative-stress and hypoxia sensitive cells than progenitor stem cells⁵⁰⁻⁵⁴, therefore more
sensitive to free radical polymerization and hypoxic conditions presented during
manufacturing. By performing this assay using HUVECs instead of progenitor cells (e.g. BM-
MSCs), masking of the level of compatibility by high resistant cells is avoid, and it can be
demonstrated with higher certainty that the biofabrication technique is cytocompatible.")

BM-MSCs cells were selected for the fabrication of SDVGs and for the immunomodulatory
study in mice, the rationale behind this is now given in line 419 (“On the other hand, bone
marrow mesenchymal stem cells (BM-MSCs) are known to have immunomodulatory activity;
therefore, it is expected that immunoreaction in presence of this cell type in the SDVG be
controlled and the effect of endotoxins ameliorated. It is important to remark that in this
study, vascular cell functionality, such as contractility, was not under evaluation, instead
functionality of BM-MSCs, known as an excellent cell source for vascular remodeling and
regeneration⁵⁶ and considered within the SDVG design.”), and reinforced in line 587 (“In vivo
immunogenicity experiments demonstrate that the encapsulation of BM-MSCs lowers the
rejection and inflammation response cause by the subcutaneous implantation of an
endotoxin-laden SDVG. The immunosuppression cell function is of importance in the design of a
new vascular graft, considering a cause of graft failure is associated to chronic inflammatory
response⁵”) and line 603 (“Additionally, mesenchymal stem cells (MSCs) can be encapsulated
as a multipotent source of the biological component with the ability to differentiate to the
desired tissues, assist in the recruitment and migration of patient’s cells by secreting different
chemokines, and help in the formation of a definitive anti-thrombogenic inner layer after
implantation⁷⁴.”).

In general terms, vascular cells were not considered in this design due to potential
immunorejection that these cells could generate in patients in any future clinical application.

In order to use autologous vascular cells, these cells need to be harvested from the same
patient. This possibility is considered risky in terms of commercial and clinical viability, as cell
harvesting and expansion would require invasive procedures, long-term cell culture and
accompanying sterility issues, making the SDVG more expensive and less applicable for urgent

situations. iPS on the other hand, are potentially an excellent source for this application due to
the low invasiveness of their harvesting procedure, however, reprogramming, expansion and
differentiation are still long and expensive procedures for this application, without mentioning
the high concern of potential teratoma formation. As identified for use in this study, donated
BM-MSCs are considered less invasive, storable and immunotolerated. Additionally, these
cells could differentiate, immunomodulate and assist tissue regeneration, being considered as
key cells in the process of remodeling and tissue integration for tissue engineered grafts.

Please describe the details of the vessel culture from days 1 to 7.

Response: This has been added and detailed in the revised version in the materials and
methods section, specifically in the cell culture sub-section in line 819 (“Cell culturing of
complete or sectioned SDVGs were performed similarly, except that culture media was
additionally supplemented with 1X amphotericin B (15290-026, Gibco, USA). This preventive
measure was performed to mitigate exposure to microorganisms post fabrication.”).

Cell survival appears to be equated to function, but it should be made clear that vessel
function (e.g. contractility) was not tested in this paper.

Response: We agree with this important comment. In the revised version, we have pointed
out in the section “Engraftment potential of the encapsulated cells in an immunocompetent
mice model” (line 414), more specifically in line 421 (“It is important to remark that in this
study, vascular cell functionality, such as contractility, was not under evaluation, instead

functionality of BM-MSCs, known as an excellent cell source for immunomodulation and
vascular remodeling and regeneration⁵⁶ and considered within the SDVG design.”), that
vascular cell functionality has not been evaluated, and that cell survival and
immunomodulatory functionality of encapsulated BM-MSCs has been measured to establish
the cytocompatibility of the manufacturing method. However, we argue that BM-MSC
function has been key in the process of remodeling and tissue fusion for other cellularized
vascular grafts. We have expanded our discussion in line 589 (“The immunosuppressive cell
function is of importance in the design of a new vascular graft, considering a cause of graft
failure is associated to chronic inflammatory response⁵. These results prove the importance of
fabricating MSC-cellularized grafts to prevent rejection and enhance the functionality of the
implant by promoting cell repopulation and definitive tissue remodeling after implantation^{74”)}
and line 603 (“Additionally, mesenchymal stem cells (MSCs) can be encapsulated as a
multipotent source of the biological component with the ability to differentiate to the desired
tissues, assist in the recruitment and migration of patient’s cells by secreting different
chemokines, and help in the formation of a definitive anti-thrombogenic inner layer after
implantation^{74”)}), arguing that BM-MSCs functionality could be a key element in the successful
grafting of the new SDVG.

SIVG does not appear to be defined anywhere in the paper

Answer: Thank you very much for pointing out this omission. This has been corrected
throughout in the revised manuscript.

Reviewer #3 (Remarks to the Author):

Title: Rapid fabrication of reinforced and cell-laden vascular grafts structurally inspired by
human coronary arteries.

Authors: Akentjew, T.L., et al.

General Comments:

This is an interesting paper describing a combination of spinning and hydrogel technologies,
that can deposit concentric layers of cell-laden hydrogels that are interspersed with re-
inforcing fibres to provide mechanical strength.

Cell layers are contained in a methacryloyl gelatin-alginate (GEAL) substrate, while the

reinforcing fibres are made from slow-degrading polycaprolactone (PCL). The idea of

interspersing mechanically reinforcing fibres within a living tissue is novel and potentially

important. However, this paper has multiple drawbacks that limit enthusiasm substantially.

Overall, important information is lacking on the process, and the figures contain many errors

and substantially lack clarity. Missing important information includes: what is the diameter of

the grafts? How long does the graft manufacturing process take – can cells survive this

period? Is the process sterile? How big are the PCL fibres? This is difficult to infer from the

paper. What is the phenotype or appearance of the cells in the construct? There is zero

histological information. How were MSC harvested and characterized? No information on this
either.

Also, there is overall a lack of care in preparation of the paper. The figure legends contain
many errors, the statistical analysis is uneven and poor overall, and the Methods descriptions
do not comport with the data in the paper. Overall, English grammar should be improved as
well.

Response: We thank the reviewer for their helpful comments. We have conducted substantial
revisions to address all of the issues pointed out. Specifically, concerning the diameter of the
graft, this information was included in line 288 of the main manuscript (“The wall thickness of
the SDVG was 0.59 ± 0.17 mm, relatively similar to the combined thickness of the middle and
outer graft layer fabricated separately, and the inner diameter 3.6 ± 0.5 mm”).

Additionally, in regard to the manufacturing time, this information is now included in line 286
(“The complete sterile SDVG fabrication (see material and methods) procedure took an
average of 30 min to manufacture once the precursor solutions were prepared”), and
discussed in line 617 (“The complete fabrication of an SDVG takes approximately 30 min.
Furthermore, this manufacturing equipment has the potential for scale-up to fabricate more
than one vascular graft at a time and can be adapted to a self-contained GMP manufacturing
unit that can be even utilized in the surgery room or in semi-centralized manufacturing
facilities close to healthcare centers. These advantages increase the feasibility of this
manufacturing method, lowering cost and manufacturing time, attractive from the
perspectives of commercialization and clinical translation.”).

Regarding the process sterility, this is now clarified in line 738 of materials and methods
through a description of precautions taken for sterility maintenance, especially for
experiments involving cell culturing or in vivo implantation, quoted here: “The whole
manufacturing process was performed under sterile conditions within a biosafety cabinet,
including the preparation of methacryloyl gelatin-alginate (GEAL) solution. After chemical
modification, and initially during dialysis, the first hour of dialysis is performed using a water
solution supplemented with 1% (v/v) chloroform for sterilization⁹². The prepared GEAL
mixture was then submitted to 3 cycles of heating and cooling (20 min at 70°C and 20 min at
4°C). Additionally, bioburden testing (Inoculation of SDVG extract in Blood agar, Sabouraud
Dextrose Agar, Valtek, Chile) and mycoplasma testing (MycoAlert™ Mycoplasma Detection
Kit, Lonza) were routinely performed for fabricated SDVG following provider instructions.
SDVG constructs used for experiments with cells or animal models were treated with 1% (v/v)
penicillin-streptomycin (15140-122, Gibco, USA) in culture media for 24 h.”.

Additionally, the information about dimensions of PCL fibre have been included in line 170
(“PCL fibre sublayers fabricated with a target of +/- 21°, had resultant average fibres angles in
the cylindrical construct of $31 \pm 31^\circ$ (Fig. 3b) in one orientation and $-28 \pm 32^\circ$ (Fig. 3c) in the
opposite orientation, with average fiber diameter of 698 ± 253 nm. The PCL fibre sublayer
fabricated targeting +/- 67°, exhibited fibre angles of $78 \pm 22^\circ$ (Fig. 3e), while the oppositely
oriented fibres were $-77 \pm 22^\circ$ (Fig. 3f), with an average fibre diameter of $1.2 \pm 0.5 \mu\text{m}$ ”).

Phenotypic appearance and histological information has been informed now in the live/dead
assay of encapsulated cells within different graft sublayer (Fig. 8) and histological analysis of

rabbit carotid grafted SDVG, respectively. The last one can be found in the new section
“Implantability study in arterial circulation using a rabbit model” in line 502 and Fig. 10.

Finally, concerning the harvesting of BM-MSCs, we specify that BM-MSCs were commercially
obtained from Lonza, and included the code and lot number in the material and methods
section, in the cell culture sub-section, line 810 (“Human umbilical cord endothelial Cells
(HUVEC) (ATCC® CRL1730™) and bone marrow derived mesenchymal stem Cells (BM-MSCs,
expanded until passage 5) (#PT-2501, Lot# 0000423370, LONZA, USA) were cultured and
expanded in high glucose Dulbecco’s Modified Eagle’s medium (DMEM) (16000-044, Gibco,
USA) supplemented with 10%(v/v) fetal bovine serum (FBS) (16000-044, Gibco, USA), 2 mM
glutamine (25030-081, Gibco, USA) and 1% (v/v) penicillin-streptomycin (15140-122, Gibco,
USA), and incubated at 37°C, 5% CO₂ and 96% of humidity.”). Lonza provided a certificate of
immunophenotypification and tridifferentiation following the ISSCR recommendations for this
type of cells.

All figure legends have been corrected, as suggested too by the reviewer N° 2, and material
and methods section improved following the reviewer’s suggestions, and addressed one-by-
one below. Regarding statistical analysis, more detailed information has been added in
material and methods (line 937: “Data are presented as mean \pm SD or mean \pm SEM, also
indicated in the figure caption. Statistical significance was determined using two-tailed Mann-
Whitney U test in compliance and cell density of bio-inspired SDVGs (Table 1, and Fig 8). Two-
tailed Mann-Whitney U test was also applied for the in vivo immunosuppression functional
experiments in the mouse model, platelet activation assay and clotting time assay (Fig 9, 10,
respectively). All statistically analyzed data comply with the non-parametric Mann-Whitney U
test. Platelet activation assay was performed utilizing 3 human blood samples from different

donors and conducted in experimental triplicates, whereas clotting time assay was done using
4 human blood donors in experimental triplicates. Sample size calculations were determine
using the on-line resource from IACUC of Boston University
([https://www.bu.edu/researchsupport/compliance/animal-care/working-with-](https://www.bu.edu/researchsupport/compliance/animal-care/working-with-animals/research/sample-sizecalculations-iacuc/)
animals/research/sample-sizecalculations-iacuc/) ^{98,99}, with a power level of 95% and
estimated variance from previous publications. Specifically for compliance experiments,
estimated variance was determined based on Claes, E.⁴⁵ and van Andel, C. J. et al.⁴⁶, whereas
for the immunocompetent mice model was based on Campos-Mora, M. et al.⁹⁶. All compared
groups are considered to have similar data distribution. In the implantability study in arterial
circulation using a rabbit model, results were presented as preliminary study, therefore only 2
animals per group were performed and no statistical analysis presented.

All experimental outcomes were obtained after blinded quantitative analysis of samples
results. For all statistically analyzed experiments, 95% of confidence was used and significance
was denoted as *P≤0.05, **P≤0.01 and ***P≤0.001. “n” values for each experiment are
informed in the figure caption.”), including as well the “n” values for each experiment and
informed in the figure caption.

Specific Comments:

1. Line 123: the statement is made that the GEAL solution is pre-crosslinked, but then the
authors state that the dipped layer is exposed to UV light to crosslink the material. please
clarify.

Response: We thank the reviewer for pointing this out. The GEAL solution was not pre-
crosslinked, we have now modified the body text to clarify this and revised Figure 2 to include
the crosslinking/photo-crosslinking step within the vessel production sequence. We have
clarified this in the manuscript in line 143: “Subsequently, the rod containing the orientated
PCL fibre sublayers was immersed (dipped) into the GEAL solution and slowly retracted whilst
spinning to allow homogenous GEAL layer photo-crosslinking using a lateral UV source (Fig.
2b).”

2. The phrase “circumferential axis” is used throughout the paper and is unclear to this
reviewer.

Response: This has now been clarified in the revised manuscript. Figure 1 has been revised
with a scheme of the multilayer vascular graft, defining the circumferential axis for clarity.

3. After spraying the PCL layers, do the fibres fuse at the points of contact? What is the data
for or against fibre fusion, which will significantly impact the vessel mechanical properties?
What is the fibre diameter, is it uniform, how is it controlled?

Response: We thank the reviewer for pointing this out. We have included more detail
concerning fibre morphologies, diameters and confirmed that the fibres are individualized by
SEM and CT investigation (Fig. 4 and 6), however, some level of fusion between fibres cannot
be discarded. We have expanded the text concerning the fibres and the implication of their
fusion on mechanical properties. We have included those comments in line 295 (“The fibres
are individualized, with minimal fibre fusion evident, according to SEM and CT (Fig. 4d and 6c,

repectively). Fibres fusion is an aspect of concern in the design of the SDVG, because fibres
fusion would indeed impact the mechanical response, mainly due to force distribution
amongst fibres at fixed contact points. Overall in the actual SDVG, free displacement of fibres
would be possible during stretching and recoil.”). Fusion of the fibres can be controlled by
fixing the distance between the SBS head (fibre emitting) and collection, if solvent evaporation
is insufficient prior to hitting the target then some fusing of the fibres can occur; we specified
the collection distance used here to obviate fibre fusion. Concerning fibre diameter, values
fluctuate between 500 nm and 1800 nm approximately depending on the angular orientation
of the SBS respect to the circumferential axis of the construct or rod. Information about the
average values and standard deviation was included in line 170 (“PCL fibre sublayers
fabricated with a target of +/- 21°, had resultant average fibres angles in the cylindrical
construct of $31 \pm 31^\circ$ (Fig. 3b) in one orientation and $-28 \pm 32^\circ$ (Fig. 3c) in the opposite
orientation, with average fiber diameter of 698 ± 253 nm. The PCL fibre sublayer fabricated
targeting +/- 67°, exhibited fibre angles of $78 \pm 22^\circ$ (Fig. 3e), while the oppositely oriented
fibres were $-77 \pm 22^\circ$ (Fig. 3f), with an average fibre diameter of $1.2 \pm 0.5 \mu\text{m}$.”). Although we
have not explored in this work the variable controlling the fibre diameters during
manufacturing, except for deposition angle (see lines 174-181), it is known from previous
research in SBS and electrospinning that solvent type and mixture and polymer concentration
are typical variable by which diameter can be controlled.

4. The legend for Figure 4 is very confusing – panels a-c are stated as having data in the grey
lines, but these lines are identical across the images, and the dotted lines are not described at

all. It is pretty impossible to tell what this figure is saying. Do the authors have data to show
on actual stress-strain curves of native human coronary arteries?

Response: We are very thankful for the reviewer comments and get feedback concerning
improvements of the figures. We have substantially revised all the figures in the manuscript
for clarity, explaining in more details the data described by the plotted lines. The legend has
been corrected too and clarified that the data of native human coronary were originated in
previous published studies.

Specifically, concerning the caption of Fig. 4, we have added the modifications and the new
caption resulted as follow: "Figure 4: Iterative improvement towards a J-shape stress-strain
curve combining wavy fibre deposition and preconditioning: (a) Optical microscope image of
a PCL fibre sublayer fabricated at a deposition angle of 21° after 1 cycle of fibre deposition
with continuous clockwise rod spinning. (b) Optical microscope image of a PCL fibre sublayer
fabricated at a deposition angle of 21° after 1 cycle of fibre deposition with alternated rod
spinning and after the preconditioning step. (c) Scanning electron microscopy image of a
series of 24 PCL fibre sublayers fabricated at a deposition angle of +/-67°. (d) Scanning
electron microscopy image of a series of 24 PCL fibre sublayers fabricated at a deposition
angle of +/-67° with alternated rod spinning and after the preconditioning step. (e)
Longitudinal strain-stress curve of a series of 24 PCL fibre sublayers fabricated at a deposition
angle of +/-21°, with and without a stretch preconditioning step of 5 cycles of
loading/unloading at 30% strain and wavy fibre deposition using the alternating rod spinning
during angled fibre deposition. The grey line represents previously published stress-strain
mechanical behavior of the media layer of the human coronary artery under longitudinal
tensile testing¹⁵."

As specified in the modified caption, human coronary data was obtained from a previous work
(Holzapfel et al (2005)), in which 13 donated human coronary arteries were evaluated. From
this data a constitutive mathematical model capable to describe the mechanical behavior in
uniaxial tensile testing was obtained. In our work, mechanical tensile testing was applied in
the same manner as their study, including strain rate as further detail in our response to
referee 2.

5. Similarly the legend for Figure 5 makes no sense – the grey lines and solid squares that are
talked about the in the legend simply aren't there in the figure. And, panels b-e contain
multiple black lines of unclear significance- if these are replicates, then should not some error
bars be included or something? In Figures 5 and 7, where is the “grey region of the range of
the model of the mechanical properties” derived from? This is not at all clear to the reader,
and serves only to confuse.

Response: All figures have been reworked, as addressed in our responses to reviewer N° 2.
Specifically concerning this reviewer's comments for Fig 5 and 7: Black lines were replicates
for the fabricated SDVG, which are expressed now as averages with their corresponding
standard deviation. The grey zone (now light green zone) corresponds to the range of values
obtained for native coronary arteries, which were extracted from previous work (Holzapfel et
al (2005)). In our work, mechanical tensile testing was applied in the same manner as this
study (Holzapfel et al (2005)), including pre-conditioning and strain rate. This was clarified in
the caption and text. Additionally, legends were included in the figures to clarify the
designation of colours and symbols. In figure 5, caption was modified and improved the clarity:

**“Figure 5: Stress-strain curves of the outer and middle graft layers based on GEAL reinforced**
**PCL sublayers. a) Iterative improvement of the middle and outer graft layers: longitudinal**
**tensile testing of the middle and outer graft layers using different numbers of middle and**
**outer graft sublayers in the construct. Stress-strain curves of the outer graft layer consisting of**
**different layer numbers. Grey dotted lines represent the average longitudinal stress-strain**
**curve of the native adventitia (closed circles) and media (closed diamonds) layer of human**
**coronary arteries¹⁵. b) and c) Longitudinal and circumferential stress-strain curve of the outer**
**graft layer composed of 5 graft (GEAL/PCL) sublayers. Green dashed lines in b and c represent**
**the average longitudinal and circumferential stress-strain curves of the native media layer of**
**coronary arteries¹⁵, respectively. d) and e) Longitudinal and circumferential stress-strain curve**
**of the middle graft layer composed of 4 graft (GEAL/PCL) sublayers. The green dashed lines**
**in d and e represent the average longitudinal and circumferential stress-strain curve of the**
**native adventitia layer of human coronary arteries¹⁵ respectively. (n=3). The shaded green**
**zones in the figures represent the range of results obtained for native human coronary**
**arteries¹⁵. Error bars = standard deviation, (n=3 independent experiments).”**

Concerning figure 7 caption was modified too and improved clarity:

**“Figure 7: Mechanical evaluation of the fabricated SDVG. a-b) Stress-strain curves of the SDVG**
**(black line) and the human coronary artery in a) longitudinal and b) circumferential stretching**
**directions (n = 5 independent experiments). The green dashed lines in a and b represent the**
**longitudinal and circumferential stress-strain curve of native human coronary arteries⁴⁶. The**
**green shaded zones represent the range of results obtained for native human coronary**
**arteries⁴⁶. (c) Cyclic tensile testing in the circumferential direction. d-f) Profiles of diameter**
**change ratio (D/D0) as function of pressure applied to the SDVG (black line, n = 5 independent**

experiments) compared with human coronary arteries (green dashed lines, n = 5 independent
experiments)^{45,46} at three different values of axial pre-stretch during testing (ez). d) ez=10% of
axial pre-stretch.e) ez=20% of axial pre-stretch. f) ez =25% of axial pre-stretch. Error bars =
standard deviation.”

6. In Table 1, what is the “n” value for these data? This bears on the statistical significance and
differences.

Response: We thank the reviewer for pointing this out. This has now been corrected in Table 1
of the revised manuscript, including details as to how the data was obtained. Specifically in
table 1 caption the following was included: “Table 1: Bio-inspired small diameter vascular graft
(n=5) and human coronary artery (n=5, data obtained from Claes, E.⁴⁵ and van Andel, C. J. et
al.⁴⁶ compliance (%C) (10-2 mmHg) at different pressure ranges and longitudinal pre-stretch
during testing (ez). Standard deviation is presented too (\pm). ”.

7. Figure 8 needs scale bars. What is the meaning of the blue color – does this signal live or
dead cells or proliferating cells? What are the dimensions of the construct? The reagent used
does not show cell replication per se – it appears to show mitochondrial activity??

Response: We have revised the manuscript and re-worked Figure 8 to include a carefully
annotated description in the caption of Figure 8. The dimensions of the graft are now clarified

in line 286 (“The complete sterile SDVG fabrication (see material and methods) procedure
took an average of 30 min to manufacture once the precursor solutions were prepared. The
wall thickness of the SDVG was 0.59 ± 0.17 mm, relatively similar to the combined thickness of
the middle and outer graft layer fabricated separately, and the inner diameter 3.6 ± 0.5 mm.
.”). Scale bars were included for the SDVG fluorescent image of SDVG in figure 8.

Additionally, the reviewer’s concern about proliferation assay and mitochondrial activity has
been clarified in the manuscript in line 380 (“In order to evaluate the cell viability after
manufacturing, a cell proliferation assay based on mitochondrial activity was performed post-
fabrication.”), and line 390 (“Nevertheless, limited diffusion of nutrients and reagents of the
proliferation kit within the graft must be taken into consideration; this could lead to an
underestimation of the mitochondrial activity, hence cell survival and proliferation”).

8. Figure 9 shows cells counts from histology, but there are no histological images shown in
the main paper. This is very disappointing.

Response: We thank the reviewer for pointing this out. Histological images of rabbit carotid
grafted SDVG are presented now in section “Implantability study in arterial circulation using a
rabbit model” (line 502) and supporting information (Fig. S8). These images correspond to
H&E staining of cellularized SDVG after 14 and 30 days post-implantation (Fig. 10h,i), and H&E
staining of acellularized (Fig S8a) and cellularized SDVG (Fig S8b) after 30 days post-
implantation. On the other hand, cell counting presented in figure 9, corresponds to cells
obtained from graft-draining lymph nodes to evaluate the level of immunoreaction. This has
been clarified in the caption and as well in line 425 (“Analysis of immunocompetent mice with

dorsal subcutaneous implantations demonstrate that SDVGs with encapsulated endotoxins
and without BM-MSCs induced graft rejection when implanted subcutaneously, characterized
by a lack of graft incision healing (see Fig. S7a), an increased number of total cells isolated
from graft-draining lymph nodes (dLNs) (mouse axillary and brachial lymph nodes), and an
increased percentage of CD4+ memory T cells and B cells in dLNs compared to BM-MSCs
cellularized graft (Fig. 9 b, c, and d respectively) ”

9. In the Methods section, the Histology paragraph talks about skin transplants and positive
controls for immunity, and talks about FACS analysis using different markers for immune cells,
but none of this data is shown in the paper. It appears as if this text was taken from another
manuscript, since it does not match up with the data in this paper?

Response: We thank the reviewer for pointing this out. This mistake has been corrected in
substantially revised materials and methods section, and as well further clarified in the
caption of Figure 9. Concerning the Figure 9, the following underlined modifications has been
included:

“Figure 9. Descriptions of immune results. (a) Schematic overview of the experimental
immune challenge to assess the immunomodulatory function of laden BM-MSC in the SDVG.
(b) Number of cells in dLN. (c) Frequency of CD62L- CD44+ cells within CD3+ CD4+ cell
population, corresponding to memory T Cells. (d) Frequency of CD19+ cells in dLN,
corresponding to B cells. (e) Frequency of CD25+ cells within CD3+ CD4+ cell population,
corresponding to activated T cells. (f) Frequency of CD62L+ CD44- cells within CD3+ CD4+ cell
population, corresponding to naïve T Cells. (g) Frequency of CD25high Foxp3+ cells within

CD3+ CD4+ cell population, corresponding to regulatory T cells. Naïve = non-operated control
mice; Suture control = operated mice without any graft; Allogeneic = operated mice with
allogeneic skin graft. Error bars = standard error of the mean. * p<0.05; ** p<0.01; ns: non-
significant. n=5 animals with 5 different SDVG fabrications. Statistical analysis was conducted
using the Mann–Whitney U test.”

Reviewers' comments:

Reviewer #1 (Remarks to the Author):

The authors are commended for performing additional SVDG strength characterization and a rabbit carotid artery interpositional implantation study. The reported strength values are reasonable albeit lower than for other TEVG that have attained large animal testing. Unfortunately, acute clotting of the grafts in the rabbit model indicates the graft, despite its noteworthy structure and compliance properties, is not yet a successful TEVG in terms of a large animal implantation; thus, the study, while comprehensive and well presented, is of limited significance given the state of the TEVG field, and I do not think its probable impact merits publication in Nature Communications.

Reviewer #2 (Remarks to the Author):

This is an exciting paper describing an innovative approach for fabrication of vascular grafts that effectively mimic the mechanical response of native vessels and provide a scaffold for vascular cells to function and remodel. The authors have effectively responded to the reviewers' recommendations, and the manuscript is substantially improved, particularly with inclusion of additional results addressing graft strength and performance in a rabbit model. There are still some grammatical issues here and there, but they are not so extensive as to lead to confusion.

One minor suggestion: Please define (in the figure caption) the angle alpha shown in Figure 1. Is it the parameter associated with the angles referenced in the caption for Fig. 1?

Ken Monson

Reviewer #3 (Remarks to the Author):

Title: Rapid fabrication of reinforced and cell-laden vascular grafts structurally inspired by human coronary arteries.

Authors: Akentjew, T.L., et al.

General Comments: This is a substantially revised manuscript regarding the use of electrospinning techniques to create PCL-gel composite tissues with mechanics similar to native artery. The changes in response to the previous review are extensive, and so this reads almost like a new paper.

Compared to the prior submission, the clarity of the procedures and the characterization of the scaffolds, particularly the mechanical characterization, are vastly improved. This reviewer is again impressed at the extent to which a biological approach to orientation of PCL fibers led to expected and predicted compliance properties. In addition, the mechanical properties of these small-diameter conduits do now appear to be compatible with arterial implantation. As such, this material characterization and the methodology combine to make this an important advance for research in the area of synthetic arterial grafts.

But while there are many improvements, there are some new weaknesses in the paper, mostly pertaining to the biological characterization of the materials. To remedy some of these deficiencies, it would be suitable (in this reviewer's opinion) to simply remove some of the confusing biological data, since it does not add significantly to the important parts of this story as it currently stands. In addition, the paper is now QUITE LONG, and should be streamlined to about half of its length, in terms of text. Specific comments below.

Specific Comments:

1. In Figure 5, providing more explicit labels on the y-axes would help readers understand which wall stress they were looking at.
2. Figure 6 does not add a lot to the paper, and could be omitted or moved to a supplement.
3. Figure 7 – again, more explicit y-axis labels, with directionality for panels a-c, and with amount of pre-stretch for panels d-f.
4. Table 1 could be omitted, as not adding a lot to the figures already presented.
5. English language correction, line 396, should read: “highly resistant cells is avoided”
6. Regarding results in lines 394-406, it should be noted that 0.02 – 0.2% survival is still very poor for HUVEC in the construct. Is poor survival the reason that HUVEC were abandoned for later implantations? Also, what was the survival of BM-MSC when implanted into the constructs?
7. The authors seem to have an incomplete appreciation of the effects of endotoxin. In lines 430-444, there is discussion of endotoxin, MSC, rejection, etc. Some corrective observations:
 - a. HUVEC will be susceptible to endotoxin – it is toxic to endothelium
 - b. Endotoxin is a strong inducer of inflammation, but not of adaptive immunity, per se. Therefore, the B- and T-cells that migrated to the implant were likely part of a non-specific inflammatory response, rather than part of an actual rejection event. If the implants did not have cells, then rejection, in the proper sense, could not have occurred.
 - c. Adding MSC to this cocktail may have resulted in fewer cells on FACS, but the meaning of this observation is really not clear.
 - d. This reviewer would recommend that the cutaneous implants be struck from the paper, since they do not add value in terms of understanding of the construct, and provide some confusing information.
8. For the rabbit carotid implants, it is worthwhile to point out that endotoxin itself can induce endothelial inflammation and hence thrombosis. This may have been why all of the implanted grafts clotted within a short time period. The amount of endotoxin in the constructs should be quantified to gain a better understanding of what is going on here.

Reviewers' comments:

Reviewer #1 (Remarks to the Author):

The authors are commended for performing additional SDVG strength characterization and a rabbit carotid artery interpositional implantation study. The reported strength values are reasonable albeit lower than for other TEVG that have attained large animal testing. Unfortunately, acute clotting of the grafts in the rabbit model indicates the graft, despite its noteworthy structure and compliance properties, is not yet a successful TEVG in terms of a large animal implantation; thus, the study, while comprehensive and well presented, is of limited significance given the state of the TEVG field, and I do not think its probable impact merits publication in **Nature** Communications.

Response: We thank the reviewer for their positive comments regarding our additional investigation to strength characterization and the rabbit carotid study.

Although we understand the author's concern pointing to the fact that the paper is not showing a successful long-term patency in a large animal model, the main focus of the manuscript is the new technology based on an innovative combination of modular and automated technologies capable to produce a tissue engineered product with standardized features, which are broadly considered essential for the efficacy of a vascular graft. We present results that show good potential towards clinical usage, specifically, platelet activation, coagulation, burst pressure, suture strength and hemostasis. Additionally, the main manuscript argues at the end of the discussion section about possible reasons that could cause the thrombus formation, basically indicating that a carotid rabbit model is not optimal for evaluating long-term patency especially for an engineered graft with mechanics and wall sizes corresponding to human coronary arteries. Therefore, it is mentioned too that only in a large animal models, the *in vivo* relevance of mechanical matching and the graft design will be fully demonstrated, which is part of our ongoing research, moving beyond the scope of the current manuscript. In regard to the possible reasons that caused the thrombus formation after 12 h, we included in the previous revision that luminal unalignment of the SDVG and the native rabbit carotid at the anastomosis zone could cause blood recirculation and stagnation, therefore, coagulation (see lines 554-556 in the new revision). In this new version, we have additionally included a second possible cause, which refers to inflammation-induced thrombogenesis due to the endotoxin levels of the SDVG (see lines 558-566 in the new revision). To complement the discussion, we quantified and reported the endotoxin level in fabricated SDVG and performed *in vivo* immunogenicity assay to evaluate the immune reaction. These results were included in the new manuscript and new supporting information, respectively, and

concluded that inflammation cannot be discarded as a possible cause of thrombus formation in the present study.

The technology presented in this manuscript is a unique rapid manufacturing process that integrate the use of natural biomaterial and cells simultaneously, conforming a disruptive strategy to overcome standardization, manufacturing control, regulatory and commercial hurdles present in the tissue engineered field, which in combination could reduce the gap for this type of product to commercialization and impact in the public health.

Reviewer #2 (Remarks to the Author):

This is an exciting paper describing an innovative approach for fabrication of vascular grafts that effectively mimic the mechanical response of native vessels and provide a scaffold for vascular cells to function and remodel. The authors have effectively responded to the reviewers' recommendations, and the manuscript is substantially improved, particularly with inclusion of additional results addressing graft strength and performance in a rabbit model. There are still some grammatical issues here and there, but they are not so extensive as to lead to confusion.

One minor suggestion: Please define (in the figure caption) the angle alpha shown in Figure 1. Is it the parameter associated with the angles referenced in the caption for Fig. 1?

Response: We thank the reviewer for his comments. We have revised the manuscript for grammatical and English errors throughout.

Additionally, we have included a definition for the angle alpha in the revised caption of figure 1, that effectively correspond to the deposition angle as stated by the reviewer.

"Figure 1: Scheme composition of the middle and outer graft layers. The middle graft layer comprises a series of four PCL/GEAL sublayers, hereafter called middle graft sublayers, with fibres deposited at angles of +/- 21° and GEAL sublayer generated after two cycles of dipping and photo-crosslinking. The outer graft layer is composed of a series of five PCL/GEAL bilayers, hereafter termed outer graft sublayers, with fibres deposited at angles of +/- 67° and GEAL sublayer generated after three cycles of dipping and photo-crosslinking. The deposition angle is represented by " α ".

Reviewer #3 (Remarks to the Author):

Title: Rapid fabrication of reinforced and cell-laden vascular grafts structurally inspired by human coronary arteries.

Authors: Akentjew, T.L., et al.

General Comments: This is a substantially revised manuscript regarding the use of electrospinning techniques to create PCL-gel composite tissues with mechanics similar to native artery. The changes in response to the previous review are extensive, and so this reads almost like a new paper.

Compared to the prior submission, the clarity of the procedures and the characterization of the scaffolds, particularly the mechanical characterization, are vastly improved. This reviewer is again impressed at the extent to which a biological approach to orientation of PCL fibers led to expected and predicted compliance properties. In addition, the mechanical properties of these small-diameter conduits do now appear to be compatible with arterial implantation. As such, this material characterization and the methodology combine to make this an important advance for research in the area of synthetic arterial grafts.

But while there are many improvements, there are some new weaknesses in the paper, mostly pertaining to the biological characterization of the materials. To remedy some of these deficiencies, it would be suitable (in this reviewer's opinion) to simply remove some of the confusing biological data, since it does not add significantly to the important parts of this story as it currently stands. In addition, the paper is now QUITE LONG, and should be streamlined to about half of its length, in terms of text. Specific comments below.

Specific Comments:

1. In Figure 5, providing more explicit labels on the y-axes would help readers understand which wall stress they were looking at.
2. Figure 6 does not add a lot to the paper, and could be omitted or moved to a supplement.
3. Figure 7 – again, more explicit y-axis labels, with directionality for panels a-c, and with amount of pre-stretch for panels d-f.
4. Table 1 could be omitted, as not adding a lot to the figures already presented.

5. English language correction, line 396, should read: "highly resistant cells is avoided"

6. Regarding results in lines 394-406, it should be noted that 0.02 – 0.2% survival is still very poor for HUVEC in the construct. Is poor survival the reason that HUVEC were abandoned for later implantations? Also, what was the survival of BM-MSc when implanted into the constructs?

7. The authors seem to have an incomplete appreciation of the effects of endotoxin. In lines 430-444, there is discussion of endotoxin, MSc, rejection, etc. Some corrective observations:

a. HUVEC will be susceptible to endotoxin – it is toxic to endothelium
b. Endotoxin is a strong inducer of inflammation, but not of adaptive immunity, per se. Therefore, the B- and T-cells that migrated to the implant were likely part of a non-specific inflammatory response, rather than part of an actual rejection event. If the implants did not have cells, then rejection, in the proper sense, could not have occurred.

c. Adding MSc to this cocktail may have resulted in fewer cells on FACS, but the meaning of this observation is really not clear.

d. This reviewer would recommend that the cutaneous implants be struck from the paper, since they do not add value in terms of understanding of the construct, and provide some confusing information.

8. For the rabbit carotid implants, it is worthwhile to point out that endotoxin itself can induce endothelial inflammation and hence thrombosis. This may have been why all of the implanted grafts clotted within a short time period. The amount of endotoxin in the constructs should be quantified to gain a better understanding of what is going on here.

Response: We thank the reviewer for their helpful comments. As suggested by the reviewer, we have moved some biological data from the main manuscript and included in the supporting information (see below for further details). Additionally, we have clarified and simplified the information obtained from the immunogenicity assays and presented in the main manuscript (complete results are included now in the supporting information). Also, we have shortened the text in about 1500 words to adjust to the length of other tissue engineering related papers previously published in Nature Communication and performed new experiment to tackle the endotoxin issue in this work. Following the specific comments, we have undertaken the following modification, addressed here comment by comment (reviewer #3 comments in blue):

Reviewer #3 comment 1. In Figure 5, providing more explicit labels on the y-axes would help readers understand which wall stress they were looking at.

- Following the reviewer's suggestions, a schematic representation of the direction of tensile testing was included in our now revised figures 3, 4, 5 and 6 for clarity.

Reviewer #3 comment 2. Figure 6 does not add a lot to the paper, and could be omitted or moved to a supplement.

- Figure 6 of the previous submitted manuscript was removed from the main manuscript as suggested by the reviewer and is now included in supporting information.

Reviewer #3 comment 3. Figure 7 – again, more explicit y-axis labels, with directionality for panels a-c, and with amount of pre-stretch for panels d-f.

- Please see our response to reviewer #3 in comment 2.

Reviewer #3 comment 4. Table 1 could be omitted, as not adding a lot to the figures already presented.

- Following the reviewer's suggestion, Table 1 has been removed and is now included in the supporting information.

Reviewer #3 comment 5. English language correction, line 396, should read: "highly resistant cells is avoided".

- After shortening the main manuscript, line 396 has been removed, therefore, the specific language correction was not necessary.

Reviewer #3 comment 6. Regarding results in lines 394-406, it should be noted that 0.02 – 0.2% survival is still very poor for HUVEC in the construct. Is poor survival the reason that HUVEC were abandoned for later implantations? Also, what was the survival of BM-MSC when implanted into the constructs?

- Although we have removed this part from the newly revised manuscript, the following discussion is necessary to be raised in order to clarify the reviewer's concern. In the previous submitted manuscript, we have attributed the low signal of metabolic activity to the limited diffusion of reagents of the proliferation kit within the graft; this could lead to an underestimation of the mitochondrial activity of cells within the graft. This is particularly true for encapsulated cells within hydrogels with limited solvent access. However, we cannot discard that the lack or decrease in the capacity of WST-1 reduction of cells is derived either from a metabolic resting induced by a free radical-derived oxidative stress during photo-crosslinking (1. Cell Cycle. 2015 Jul 3; 14(13): 2022–2032 2. Ann Biomed Eng. 2017 Feb; 45(2): 360–377. 3. Gene 337 (2004) 1 – 13), or by NADH depletion in response to higher concentration of free radicals and hypoxia after photo-crosslinking (1. FASEB J. 2009 Sep; 23(9): 3159–3170. 2. Free Radic Biol Med.

2015 Feb; 0: 281–291), or simply by an interference of the photoinitiator-derived free radical with the reduced form of the 1-Methoxy-5-methylphenazinium methyl sulfate during the electron transfer in the reduction process of WST-1 to formazan. Although these phenomena could occur in conjunction and affect the reduction of WST-1 during the assay, viability measurement based on assays that evaluate membrane integrity still indicate that cells are viable (please see Figure 7 of the new manuscript). Considering the mentioned doubts about the proficiency of this method in measuring cell viability in the actual cell-encapsulation scenario, and the fact that the cell viability and compatibility of cells within GelMA hydrogels has been previously explored by this group (Biofabrication. 2016 Dec 1;9(1):015001) and other research groups (Biomaterials. 2010 Jul;31(21):5536-44), we have decided to include only live/dead assay and immunosuppression functionality in the *in vivo* model as proofs of viability of cells in the present manuscript.

Concerning the decision of abandoning HUVECs in the final SDVG design, this is not related to the survival of HUVECs but to the expected and more appropriate role of BM-MSCs in the final design with potential regenerative activity after implantation. Reasonings in this regard are included in the new revision:

In lines 363-383:

"Having considered the incorporation of a cellular component essential for the design and successful outcome in a transplantation scenario^{32,49,50}, important practical implication must be taken into account when choosing an appropriate cell type. Although autologous vascular cells are the preferable choice, invasive harvesting and long-term culturing, specially for elderly patients, make this option risky in terms of commercial and clinical viability. Induced pluripotent stem cells on the other hand, are potentially an excellent source for this application⁵¹ due to the low invasiveness of their harvesting procedure and autologous nature, however, reprogramming, expansion and differentiation are still a long and expensive procedures, and the frequency of point mutations⁵² has generated serious concern about the safety of these cells. Allogenic bone marrow mesenchymal stem cells (BM- MSCs) are considered less invasive, storable, economically feasible, immunotolerated, and have been physiologically implicated in vascular repair and remodeling⁵³. Additionally, BM- MSCs are known to have immunomodulatory activity; therefore, it is expected that immunoreaction or inflammation in presence of this cell type in the SDVG would be controlled or ameliorated. In this regard, BM- MSCs has been proposed in this study as a source of biological function for the actual SDVG design. Analysis of

immunocompetent mice with dorsal subcutaneous implantations has demonstrated that SDVGs with encapsulated BM-MSCs are capable to control an inflammatory response, whereas non-cellularized SDVG not (see Fig. S9, S10 and supporting information for further details). This demonstrates also that cells maintain their viability and functionality after being subjected to the manufacturing process. It is important to remark that in this study, vascular cell functionality, such as contractility, was not under evaluation, instead functionality of BM-MSCs, known as an excellent cell source for immunomodulation, vascular remodeling and regeneration⁵⁴.

In regard to the BM-MSCs survival, quantification of live cells was included in the new manuscript in lines 340-344: "Using this time bone marrow-derived mesenchymal stem cells (BM-MSCs), and a LIVE/DEAD® cell staining assay, viability analysis of cells located within different sublayers of the fabricated SDVG was performed at different time points of static cell culturing (Fig. 7f-h). Evaluation on day 7, 14, 21 and 28 resulted in 71, 84, 87 and 92% viability respectively (Fig. S8). These results confirm cells survival and even proliferative capacity on day 28 post fabrication (Fig 7i)".

Reviewer #3 comment 7. The authors seem to have an incomplete appreciation of the effects of endotoxin. In lines 430-444, there is discussion of endotoxin, MSC, rejection, etc. Some corrective observations:

Considering the reviewer's suggestion, the subcutaneous implantation was moved to the supporting information and taken only as an additional information to prove the immunosuppressive capacity of BM-MSCs in the present construct. Additionally, the used of a low endotoxin alginate for the SDVG construct was specified in the materials and methods section in lines 593-594 and 600-602, clarifying too that the use of an alginate with higher level of endotoxin was only for the subcutaneous assay in the context of testing the immunosuppressive activity of encapsulated BM-MSCs.

Reviewer #3 comment 7a. HUVEC will be susceptible to endotoxin – it is toxic to endothelium.

Comment 7a: Although the LPS toxicity on endothelial cells has been previously established by other authors (Infect Immun. 1993 Aug; 61(8): 3149–3156.), in this investigation, low endotoxin alginate has been used in the fabrication of SDVG, as mentioned previously, therefore, the authors of this article did not consider the possible toxicity of endothelial cells after the SDVG implantation. However, discussion and experimental proofs regarding this point are mentioned in the paper and in this document further below.

Reviewer #3 comment 7b. Endotoxin is a strong inducer of inflammation, but not of adaptive immunity, per se. Therefore, the B- and T-cells that migrated to the implant were likely part of a non-specific inflammatory response, rather than part of an actual rejection event. If the implants did not have cells, then rejection, in the proper sense, could not have occurred.

As well mentioned by the reviewer, and due to the nature of the implanted acellular SDVG, effectively, we should not talk about rejection, therefore, we have changed this concept for inflammation and wound healing, when it comes to consider evaluations of the faster incision closure for cellularized SDVG. These changes are listed below:

- In lines 375-382 of the main manuscript: "Analysis of immunocompetent mice with dorsal subcutaneous implantations has demonstrated that SDVGs with encapsulated BM-MSCs are capable to control an inflammatory response, whereas non-cellularized SDVG not (see Fig. S9, S10 and supporting information for further details). This demonstrates also that cells maintain their viability and functionality after being subjected to the manufacturing process. It is important to remark that in this study, vascular cell functionality, such as contractility, was not under evaluation, instead functionality of BM-MSCs, known as an excellent cell source for immunomodulation, vascular remodeling and regeneration⁵⁴."
- In Supporting information, subsection "Results of immunosuppressive activity of the encapsulated cells in the SDVG", the following has been included: "Analysis of immunocompetent mice with dorsal subcutaneous implantations demonstrate that SDVGs with encapsulated endotoxins and without BM-MSCs induced a graft-derived inflammatory reaction, characterized for instance by a delayed graft incision healing (see Fig. S9a), an increased number of total cells isolated from graft-draining lymph nodes (dLNs) (mouse axillary and brachial lymph nodes), and an increased percentage of CD4+ memory T cells and B cells in dLNs compared to BM-MSCs cellularized graft (Fig. S10 b, c, and d respectively). Whereas subcutaneously implanted SDVGs with encapsulated BM-MSCs, exhibited no signs of inflammation and the incision healed after 14 days (Fig. S9b). Reduced cell numbers in dLNs, and an augmented percentage of activated CD4+ T cells, CD4+ naïve T cells, and CD4+ regulatory T cells (Fig. S10 e, f and g respectively) compared to implanted acellularized grafts was also found. These results, and considering the immunophenotypic data of allogeneic skin graft, known for conducting inflammation and immune rejection^{7,8},

suggests that an immunomodulation is being carried out by the viable and functional encapsulated BM-MSCs, mainly characterized by an immunotolerance of the endotoxin-laden graft⁹, decreased presence of B-cells¹⁰ and increased presence of regulatory T cells in dNLS¹¹, all previously described as functions of BM-MSCs."

In regard to the increase of T-cells and B-cells induced by the presence of endotoxin in the subcutaneously implanted SDVG, we agree with the reviewer about the non-specific inflammatory response as the main cause of increased number of immune cells. In the succession of event after incision, it has been reported that within the first week, recruitment of antigen-presenting cells, T-cells (J Immunol. 2010 May 15; 184(10): 5423–5428.) and B-cells (Wound Repair Regen. 2017 Sep; 25(5): 774–791) may occur as a non-specific respond. However, LPS is considered a strong adjuvant with implication in clonal expansion of T-cells for example (Crit Rev Immunol. 2008; 28(4): 281–299), either derived from presentation of SDVG biomaterial antigens or unknown antigen expressed on damaged keratinocytes (J Immunol. 2010 May 15; 184(10): 5423–5428.). Nevertheless, these considerations are included in the new manuscript, describing the phenomenon more as a non-specific inflammatory reaction and not as an implant rejection.

Reviewer #3 comment 7c. Adding MSC to this cocktail may have resulted in fewer cells on FACS, but the meaning of this observation is really not clear.

Although we have move these results and discussion to supporting information, we have included an explanation in supporting information referring to the immunomodulatory effect of BM-MSCs in the inflammatory reaction triggered by the implantation of endotoxin-laden SDVG:

".....suggests that an immunomodulation is being carried out by the viable and functional encapsulated BM-MSCs, mainly characterized by an immunotolerance of the endotoxin-laden graft⁶, decrease in the number of dLN cells, decreased presence of B-cells⁷ and increased presence of regulatory T cells in dNLS⁸, all previously described as functions of BM-MSCs....."

Additionally, we have simplified the message of these results in the main manuscript in order not to deviate the focus of the study:

In lines 375-382: "Analysis of immunocompetent mice with dorsal subcutaneous implantations has demonstrated that SDVGs with encapsulated BM-MSCs are capable to control an inflammatory response, whereas non-cellularized SDVG not (see Fig. S9, S10 and supporting information for further details). This demonstrates

also that cells maintain their viability and functionality after being subjected to the manufacturing process. It is important to remark that in this study, vascular cell functionality, such as contractility, was not under evaluation, instead functionality of BM-MSCs, known as an excellent cell source for immunomodulation, vascular remodeling and regeneration⁵⁴.

In lines 516-520: "*In vivo* immunogenicity experiments (Fig. S10) demonstrate that the encapsulation of BM-MSCs lowers the inflammation response potentially caused by the implantation. The immunosuppressive cell function is of importance in the design of a new vascular graft, considering that a cause of graft failure is associated to chronic inflammatory response⁵"

Reviewer #3 comment 7d. This reviewer would recommend that the cutaneous implants be struck from the paper, since they do not add value in terms of understanding of the construct, and provide some confusing information.

This section was moved to the supporting information and referred in the main manuscript as an experimental proof of the functionality of encapsulated BM-MSCs and for the possible role in controlling inflammation after implantation. Please see responds to comments 7c.

Reviewer #3 comment 8. For the rabbit carotid implants, it is worthwhile to point out that endotoxin itself can induce endothelial inflammation and hence thrombosis. This may have been why all of the implanted grafts clotted within a short time period. The amount of endotoxin in the constructs should be quantified to gain a better understanding of what is going on here.

The following discussion was included in the discussion section in relation to the possible cause of thrombus formation:

In lines 545-573: "Although patency was only observed after a 12 h period (Fig 8), the new manufactured SDVG has proven suitable in terms of maximal burst pressure (Fig S6), suture retention (Fig S7), hemocompatibility and blood leak-proof grafting demonstrated by *in vivo* rabbit models (see Fig 8). The rabbit model was applied here as it has been considered adequate for studies of small diameter vascular conduits due to the good similarity in thromboplastic and fibrinolytic properties with humans⁷⁸. Although the diameter of the coronary-like SDVG was adjusted in this study to test the grafting capability in a carotid rabbit model, graft wall thickness and mechanical properties were tailored towards the human coronary artery. Suturing conduits of equivalent diameters but with dissimilar wall thickness, could result in luminal unalignment between the graft and anastomosed natural vessel (see Fig. 8d). This is especially true for 1.5 mm conduits in diameter,

even for a highly skilled and trained vascular surgeon. Unaligned luminal edges generate protruding obstacles for the laminal blood flow vectors at the anastomosis, creating recirculation and stagnation zones. According to previous studies, recirculation and stagnation points intensify the thrombus formation⁷⁹. Although still conjectural, this could be the reason of short patency of SDVG in the rabbit carotid model, considering that even for ePTFE vascular prostheses, patency in this model is retained for longer than 1 week⁸⁰. Another possibility could be associated to exacerbated inflammatory reaction and acute thrombogenesis⁸¹ after implantation of the SDVG and triggered by the presence of traces of endotoxin in alginate or GelMA⁸². Notwithstanding that GelMA was carefully prepared and a low endotoxin alginate selected, SDVG constructs were submitted to endotoxin level quantification and *in vivo* immunogenicity study utilizing the complete SDVG constructs and its individual components (GelMA, alginate, PCL) (see Fig. S12). Although the endotoxin level present in the SDVG (3.11 EU/ml), which is in the range that a previous study reported induction of inflammatory reaction in macrophages⁸², the *in vivo* immunogenicity results showed a low immune reaction for all individual components of the SDVG and the complete SDVG (Fig S12). Therefore, the conclusion that the short-term patency due to thrombus formation is in response to an exacerbated immunoreaction cannot be discarded. In this regard, use of endotoxin-free biomaterials and additional endotoxin control strategies will be required for further translational potential.

Although long-term grafting evaluation is required to demonstrate the efficacy of the bio-inspired SDVG design, these preliminary results show good potential towards clinical usage. Next steps in the development of this SDVG would certainly demand the use of larger and clinically relevant animal models with experimental follow up longer than a year^{68,83,84}.

In this regard, and according to the previous quoted section, we have included the presence of endotoxin as a possible cause of thrombogenesis, and also measured the level of endotoxin in the SDVG construct. Additionally, we have correlated this level of endotoxin with *in vivo* results in which individual material components and the complete SDVG, including the low endotoxin alginate, were tested for immune reaction in the same subcutaneous mice model, not clearly showing the exacerbated increment of lymphocytes counting in dLN as observed in SDVG fabricated with alginate with higher level of endotoxins.

REVIEWERS' COMMENTS:

Reviewer #3 (Remarks to the Author):

The authors have now addressed all of the comments of this reviewer, and the manuscript has been suitably revised.